# Brain criticality predicts individual levels of inter-areal synchronization in human electrophysiological data

Marco Fuscà [1,2,10], Felix Siebenhühner [2,3,10], Sheng H. Wang [2,4,5,6,10], Vladislav Myrov[4], Gabriele Arnulfo [2,7], Lino Nobili [8,9], J. Matias Palva[1,2,4,11] & Satu Palva [1,2,11] ✉

Neuronal oscillations and their synchronization between brain areas are fundamental for healthy brain function. Yet, synchronization levels exhibit large inter-individual variability that is associated with behavioral variability. We test whether individual synchronization levels are predicted by individual brain states along an extended regime of critical-like dynamics – the Griffiths phase (GP). We use computational modelling to assess how synchronization is dependent on brain criticality indexed by long-range temporal correlations (LRTCs). We analyze LRTCs and synchronization of oscillations from resting-state magnetoencephalography and stereo-electroencephalography data. Synchronization and LRTCs are both positively linearly and quadratically correlated among healthy subjects, while in epileptogenic areas they are negatively linearly correlated. These results show that variability in synchronization levels is explained by the individual position along the GP with healthy brain areas operating in its subcritical and epileptogenic areas in its supercritical side. We suggest that the GP is fundamental for brain function allowing individual variability while retaining functional advantages of criticality.

Transient, long-range synchronization of neuronal oscillations regulates neuronal processing and communication in large-scale neuronal circuits, which is essential for cognitive functions and behavior[1–6]. Healthy brains operate with moderate levels of synchronization, while inadequate or excessive synchronization is an endophenotype in several brain disorders and contributes to the functional deficits therein[7–9]. Even among healthy subjects, however, there is considerable variability in mean synchronization levels of large-scale brain networks across individuals, which is associated with interindividual variability in cognitive performance[10–16].

The framework of brain criticality offers an approach to understanding variability in brain dynamics. The 'critical brain' hypothesis posits that neuronal systems in vivo have an operating point at the critical transition between subcritical and supercritical phases in the system's state space[17–20] (Fig. 1a). Such an operating point at the edge of chaos[21,22] would yield as observables emergent synchronization dynamics with large variance and scale-free and long-range spatio-temporal correlations[23,24] (see Fig. 1a). Critical dynamics are characterized by a balance between attenuating and amplifying activity propagation and moderate levels of synchronization that are

[1]Centre for Cognitive Neuroimaging, School of Psychology and Neuroscience, University of Glasgow, Glasgow, UK. [2]Neuroscience Center, HiLIFE-Helsinki Institute of Life Science, University of Helsinki, Helsinki, Finland. [3]BioMag Laboratory, HUS Medical Imaging Center, University of Helsinki, Aalto University, and Helsinki University Hospital, Helsinki, Finland. [4]Department of Neuroscience and Biomedical Engineering, Aalto University, Espoo, Finland. [5]CEA, NeuroSpin, Gif-sur-Yvette, France. [6]MIND team, Inria, Université Paris-Saclay, Bures-sur-Yvette, France. [7]Dept. of Informatics, Bioengineering, Robotics and System engineering, University of Genoa, Genoa, Italy. [8]Child Neuropsychiatry Unit, IRCCS, Istituto G. Gaslini, Department of Neuroscience (DINOGMI), University of Genoa, Genoa, Italy. [9]"Claudio Munari" Epilepsy Surgery Centre, Niguarda Hospital, Milan, Italy. [10]These authors contributed equally: Marco Fuscà, Felix Siebenhühner, Sheng H. Wang. [11]These authors jointly supervised this work: J. Matias Palva, Satu Palva. ✉e-mail: satu.palva@glasgow.ac.uk

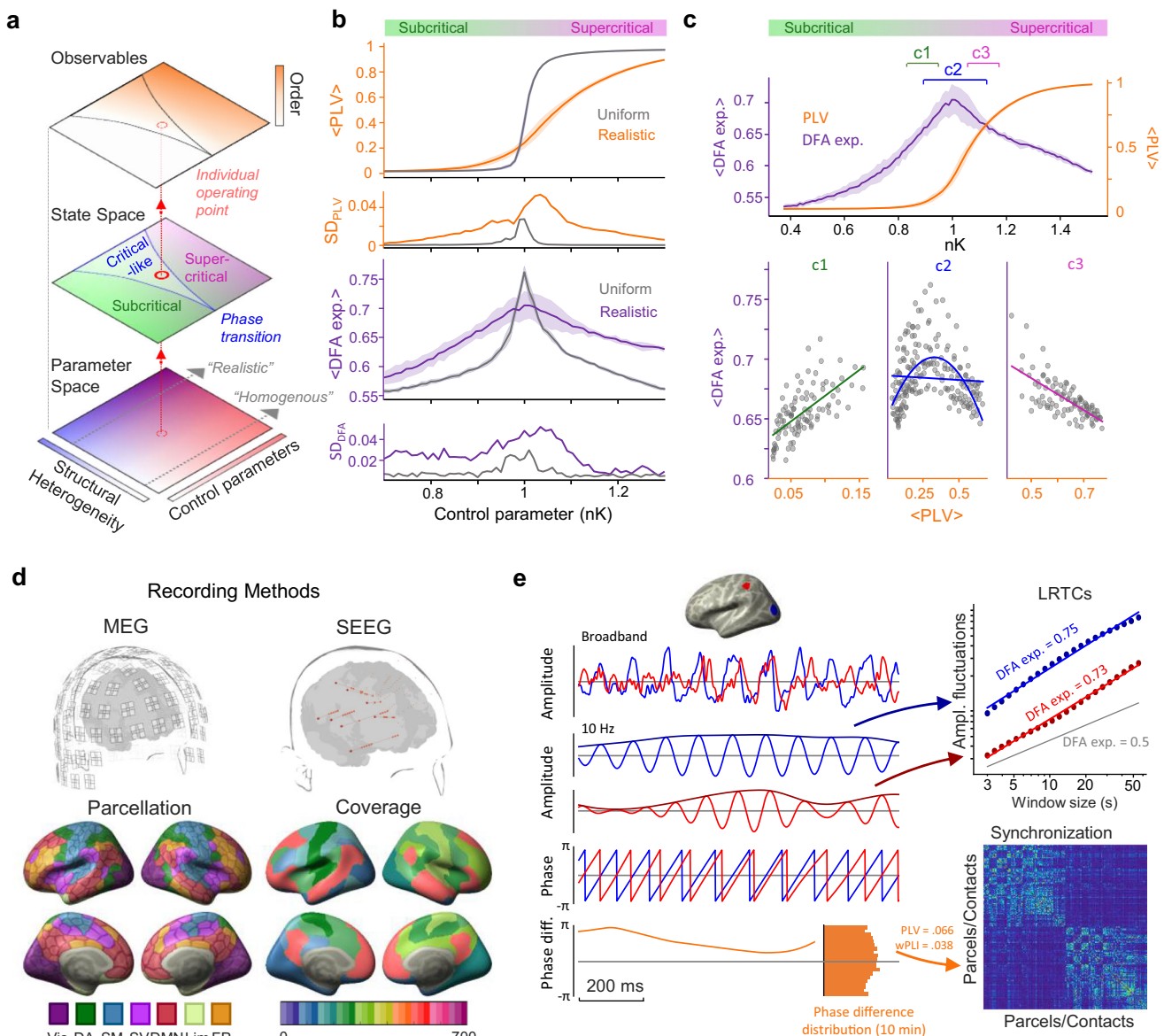

**Fig. 1 | Study schematics and modeling the co-emergence of LRTCs and inter-areal synchronization. a** Concept of critical-like brain dynamics. The state-space of critical dynamics emerges from a position in parameter space, leading to observables such as synchronization and long-range temporal correlations (LRTCs). **b** Computational modeling of critical dynamics. Hierarchical Kuramoto models with either uniform or realistic heterogenous connection weights for inter-node coupling were used to simulate classical criticality or Griffiths phase (GP). Mean phase-locking value (PLV) and scaling exponents from detrended fluctuation analysis (DFA, indexing LRTCs) are shown (with SD) as a function of the normalized control parameter (nK), the coupling strength between oscillators within each node. The model shows an exponential increase in PLV with uniform weights, and a nearly linear increase with realistic connection weights. DFA exponents peak at the critical point where nK = 1, but the peak is wider and the standard deviation larger in the realistic GP model. **c** The differential relationships of mean PLV and DFA (shown with SD) of nK for realistic connection weights simulating GP. PLV increases monotonically with K, whereas the DFA exponent peaks around nK = 1. **c1**–**c3** Scatterplots of PLV and DFA exponent from the subcritical, critical, and

supercritical sides, respectively. **c1** In the subcritical side, the relationship between PLV and DFA is well-described with a positive linear fit (green line). **c2** Around the critical point, the relationship of PLV and DFA is better described with a concave quadratic fit (dashed blue line) than a linear fit (solid). **c3** In the supercritical side, the relationship is best described with a negative linear fit (magenta line). **d** Left: Magnetoencephalography (MEG) data was source-reconstructed and parcellated into Schaefer atlas. Right: Example of stereo-encephalography (SEEG) intracranial electrode layout (top) and cohort-wide contact coverage per functional subsystem in Schaefer atlas (bottom). **e** Time series data (top left) were filtered to obtain narrow-band amplitudes (2nd−3rd rows), phase time series (4th row), and phase difference time series (bottom row). Amplitude time series were used to compute DFA exponents (top right). Pairwise phase synchronization with PLV or weighted phase-lag index (wPLI) between all brain areas (bottom right) was computed from phase difference time series. Source data for panels b,c is provided in the source data file. Functional subsystems: Vis visual, DA dorsal attention, SM somatomotor, SV salience and ventral attention, DMN default mode network, Lim limbic, FP frontoparietal.

functionally neither inadequate nor excessive[2,22,25,26]. Operation at criticality endows a system with many functional benefits, such as maximal dynamic range, information transmission, and representational capacity, all of which are instrumental to healthy brain function[27,28]. The operating point (i.e., the system's position in state space) is

dependent on the system's position in the parameter space (see Fig. 1a). In neuronal systems, the excitation-inhibition (E/I) ratio is thought to be the primary control parameter[29] so that perfectly balanced E/I leads to operation at the critical point. Excessive inhibition leads to operation in the subcritical regime where neuronal

signaling is attenuated, and spatiotemporal correlations are exclusively short-ranged. Excessive excitation leads to supercritical dynamics with escalating, self-amplifying neuronal activity that propagates across the system[21]. As a hallmark of brain criticality, in electrophysiological data, local oscillations also demonstrate scale-free long-range temporal correlations (LRTCs), i.e., power-law autocorrelations in amplitude fluctuations across lags of hundreds of seconds[30–33] and neuronal avalanches that are power-law scaled cascades of neuronal activity propagating across the neocortex in both microscopic[34] and macroscopic[32,33,35] scales of brain networks.

However, despite extensive lines of research on inter-areal synchronization and critical brain dynamics, there is only sparse evidence linking the strength of inter-areal correlations with individual critical dynamics in the human brain[36] and no experimental papers addressing inter-areal synchronization of oscillations in this context. Moreover, recent theoretical studies suggest that because of diverse structural and mechanistic heterogeneities, neuronal systems are unlikely to operate at a singular critical point[37,38]. Instead, neuronal systems have been proposed to operate in an extended regime of critical-like dynamics known as the Griffiths phase (GP)[38–40].

A GP is characterized by power laws extending over broad regions in parameter space and thus the stretching of a critical point into a wider critical regime (Fig. 1b). In modeling studies, the intrinsic heterogeneity in the white-matter structural connectivity linking human brain areas lead to the emergence of a GP[38,39]. While there is only little experimental brain data to support this hypothesis, it is in line with findings showing that different brain systems exhibit partially independent operating points[16]. The notion of GP in brain dynamics thus implies that instead of there being a single critical point that yields optimal brain functioning, there exists a wider range of possible operating points, all of which yield critical-like dynamics and consequential functional benefits.

Here, we build on the framework that healthy brains operate in an extended critical regime, the GP, and posit that the operating points would vary near a single critical point both across individuals and across neuronal subsystems within individuals. We thus hypothesized that individual variability in the operating points within the critical regime would predict the individual co-variability in the synchronization levels and LRTCs (Fig. 1c). If healthy brains operate in a mostly subcritical regime as suggested previously[41], the individual levels of inter-areal oscillatory synchronization should be positively correlated with LRTCs, both across individuals as well as across brain areas. We probed both meso- and macro-scale ongoing neuronal dynamics by using invasive intra-cerebral stereo-EEG (SEEG) and non-invasive magnetoencephalography (MEG) recordings, respectively, and quantified the strength of LRTCs and inter-areal phase synchronization in large cohorts of subjects.

## Results

We first assessed the emergence of critical dynamics and stretching of the critical point into a Griffiths phase (GP) in a generative model of synchronization dynamics (see Fig. 1a, b). The model was then used to derive predictions about the correlations between LRTCs and long-range synchronization and how these correlations are dependent on the control parameters (see Fig. 1c) that regulate the operating point in state space (Fig. 1a). Second, we analyzed resting-state brain activity recorded from healthy subjects with non-invasive magnetoencephalography (MEG) and from subjects with drug-resistant epilepsy with intracranial stereo-electroencephalography (SEEG; Fig. 1d). We then assessed the presence of LRTCs in amplitude fluctuations using detrended fluctuation analysis (DFA) within all regions and estimated the pairwise phase synchronization between all cortical regions (Fig. 1e). Lastly, we estimated whether the strength of synchronization across the whole-brain and individual cortical regions were correlated with DFA exponents across individuals in line with modeling predictions.

## Computational modeling reveals the co-variability of neuronal synchronization and LRTCs in GP

To assess how large-scale synchronization and local LRTCs depend on the operating point, we used a hierarchical variant[15,42] of the Kuramoto model[43] to simulate emergent local and large-scale dynamics. The hierarchical model comprised 400 nodes corresponding to the brain regions of the Schaefer atlas, with each node modeled as a Kuramoto model of 500 all-to-all connected oscillators. A within-node control parameter K was used to scale the uniform, all-to-all coupling among the oscillators in each node, while an independent inter-node control parameter L scaled the structural connectivity between the nodes. We first simulated both classical brain criticality and critical-like GP by varying the heterogeneity of structural connectivity (see Parameter space in Fig. 1a). To this end, for classical brain criticality, all nodes were coupled with uniform connectivity, while for the GP, the coupling between nodes was defined with the realistic and highly heterogenous structural connectome of the human cerebral white-matter tracts. We assessed inter-areal synchrony using the phase-locking value (PLV) and the LRTCs using the scaling exponent obtained with detrend fluctuation analysis (DFA) across a varying K value with ten virtual subjects with random structural variability for each K (see Methods).

Both uniform and realistic structural connectomes gave rise to a state space comprising subcritical (low synchrony, measured with PLV) and supercritical (high synchrony) phases with a smooth second-order phase transition in between with emergent power-law dynamics (strong LRTCs, measured with DFA) peaking at this transition (Fig. 1b), in line with a large body of prior findings[15,44]. For realistic structural connectivity, however, the transition was significantly wider than for the uniform, as depicted in the schematic Fig. 1a and reproducing the fundamental prior findings[37]. Note that because of the limited size of the model and structural variability, the uniform condition does not exhibit a point-like critical transition but a narrow regime. Importantly, while the structural variability among the virtual subjects was identical between realistic and uniform conditions, the resulting variability (see SD in Fig. 2b) in measures of dynamics was much greater for realistic connectivity, implying that the GP not only extends the range of operating points that yield critical-like dynamics but also amplifies the impact of individual structural variability on emergent dynamics.

We then used three cohorts of the virtual subjects to assess how long-range synchrony and LRTCs covary across the extended critical regime (Fig. 1c): for subjects in the subcritical side of a GP, the DFA exponents and PLV are positively correlated (c1), while this correlation is negative for subjects in the supercritical side (c3). Finally, a cohort of subjects at the peak of a GP will exhibit a negative quadratic correlation (inverted-U shape) between PLV and DFA. This modeling work thus shows both how structural heterogeneities stretch the critical point into an extended regime of critical-like dynamics and how the correlations of synchronization and LRTCs can be leveraged to infer the range of operating points exhibited by cohorts of subjects.

## Synchronization and LRTCs are positively correlated across individuals in MEG and SEEG

To test empirically this hypothesis and model predictions on individual brain dynamics in MEG and SEEG resting-state recordings (see Fig. 1d), we quantified LRTCs with DFA[30] and pairwise, all-to-all synchronization with the weighted phase-lag index (wPLI, in MEG) and PLV (in SEEG) (see Fig. 1e). We first assessed narrow-band oscillatory brain dynamics at the whole-brain level (see Methods). We used graph strength (GS, average of all pairwise synchronization estimates) to assess the individual mean level of global synchronization, and the mean DFA scaling exponents across the cortical surface to assess global LRTCs. In MEG ($N = 192$ sessions from 52 subjects), the grand-average graph strength GS peaked in the alpha frequency band (8–14 Hz), while in SEEG ($N = 57$ patients, one session each) GS peaked slightly lower in the theta-alpha bands (range 5–10 Hz; Fig. 2a), in line

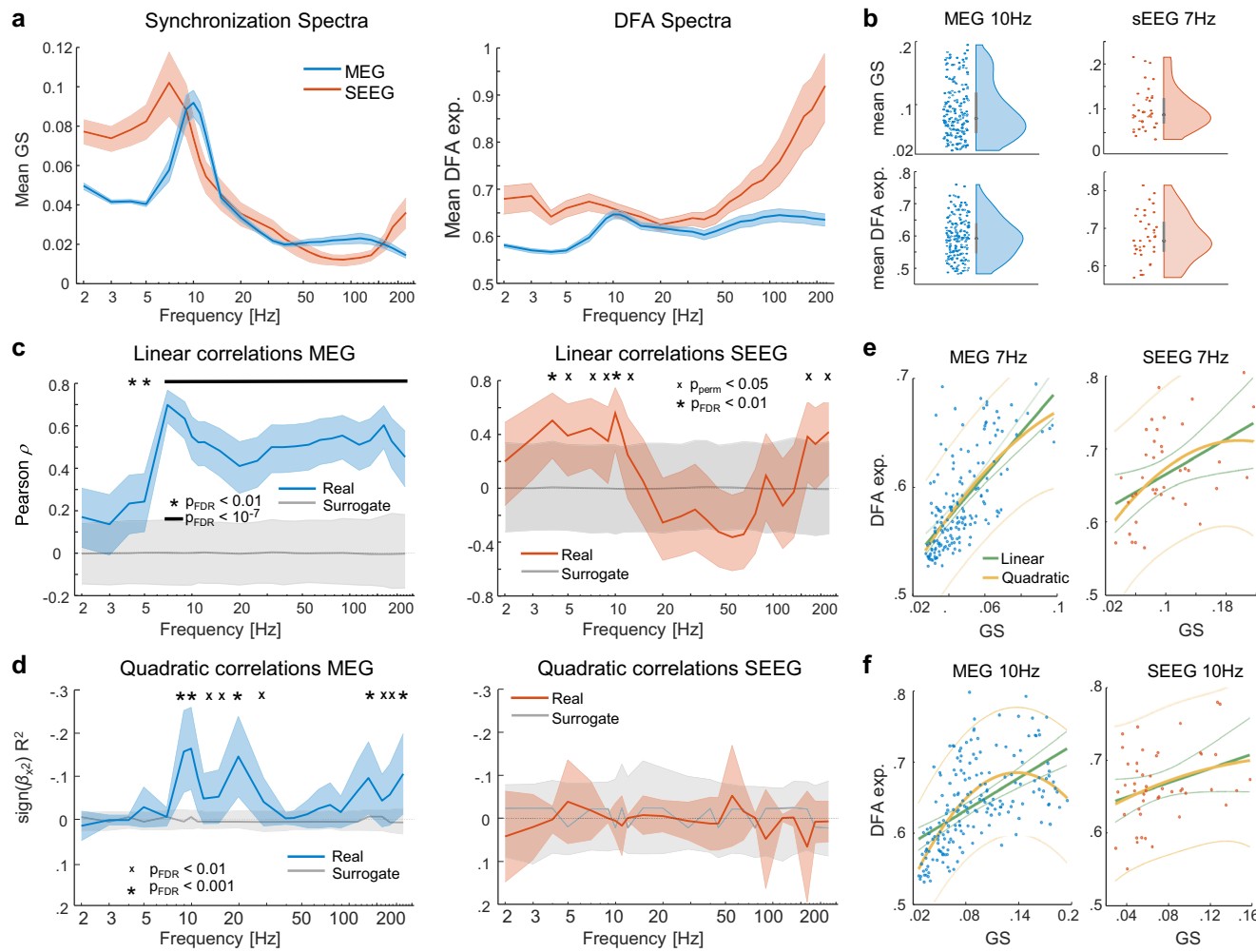

**Fig. 2 | Correlations of global synchronization and LRTCs. a** Grand-average graph strength (GS, left) and detrended fluctuation analysis (DFA) exponents (right) in MEG and SEEG data as a function of frequency. Shaded areas represent 95% bootstrapped confidence intervals. **b** Distributions for GS (top) and DFA exponents (bottom) in MEG at 10 Hz (left, $N = 192$ recordings) and in SEEG at 7 Hz (right, $N = 57$ recordings), each dot representing one dataset, bars denoting median and 25th and 75th percentile. **c** Mean linear correlations (Pearson's $\rho$) of GS and DFA as a function of frequency. Real correlations are shown in blue/red, and surrogate means in gray, with the shaded areas representing 95% confidence intervals. The asterisks at the top indicate $p_{FDR} < 0.01$ (case-resampling permutation test, two-sided, significant after FDR correction with Benjamini–Hochberg), the black line at

the top indicates $p_{FDR} < 10^{-7}$, and "x" indicates $p_{perm} < 0.05$. **d** Partial-quadratic correlations (after the linear component has been partialed-out) of GS and DFA exponent are indexed by the $R^2$ times the sign of the quadratic beta (notice the y-axis is reversed, with negative on top). Shaded errorbars and the surrogates are obtained as in **c**. Asterisks indicate $p_{FDR} < 10^{-3}$, (correction with Benjamini–Hochberg), "x"s indicate $p_{FDR} < 0.01$. **e, f** Scatterplots of GS and DFA at peak correlation frequencies 7 and 10 Hz, each dot representing one dataset. Fits are shown as solid lines, green for linear and yellow for quadratic, with the faint dotted side lines showing the 95% prediction bounds. Source data for panels **a–f** is provided in the source data file.

with previous studies showing a shift from the alpha to theta in SEEG[42,45]. In MEG, the grand-average DFA showed a well-delineated alpha-band peak, as seen in GS above, and a broader gamma-band (50–100 Hz) peak as well. In SEEG, the DFA peaked in the theta-alpha range (5–10 Hz) as well as in the delta-band (2–4 Hz) and exhibited a monotonic near-linear increase in gamma frequencies (Fig. 2a). Plotting of individual values for each set demonstrated clearly that both GS and DFA values exhibited very large interindividual variability across subjects (Fig. 2b).

To test whether interindividual variability in neuronal synchronization could be explained by brain criticality, and particularly by the putative individual operating points along an extended critical regime, we investigated whether GS and DFA values would be correlated and co-vary across subjects. In MEG, the positive correlations between GS and mean DFA exponent were significant in all frequencies above 4 Hz (Pearson correlation test, $N = 192$, FDR-corrected, $p_{FDR} < 0.01$ for 4–5 Hz and $p_{FDR} < 10^{-7}$ for the higher frequencies, reaching $p_{FDR}$

$< 10–25$ at 7 Hz; Fig. 2c). In SEEG, the correlations between GS and mean DFA were weaker than in MEG in higher frequencies, but significant in the theta-alpha (Pearson correlation test, $N = 57$, range 4–12 Hz, $p_{FDR}$ $< 0.01$ for 4 and 10 Hz) and ripples-gamma (>165 Hz) bands (Pearson correlation test, $N = 57$, $p_{perm} < 0.05$, Fig. 2c). In SEEG, correlations between GS and DFA values were negative, but non-significant, in beta and gamma bands (range 20–60 Hz).

Since our modeling results showed that subjects operating around the peak of the critical regime would exhibit a quadratic correlation in addition to linear ones, we also estimated quadratic trends and their direction with partialed-out linear influences between GS and mean DFA exponent, using the R2 regression statistic multiplied with the sign of the quadratic coefficient. In the critical regime, the quadratic coefficient should be negative, denoting a peak, i.e., a concave curve with an inverted-U shape. Significant correlations with a negative quadratic coefficient between GS and mean DFA were indeed observed in MEG data in alpha (8–14 Hz, peak frequency at 10 Hz with

$p_{FDR} < 10^{-6}$), beta (15–29 Hz) and ripples-gamma (>135 Hz; Fig. 2d) bands. In SEEG data, we found no evidence for significant quadratic correlations between GS and mean DFA exponent in any frequency (Fig. 2d). As a post hoc visualization of these findings, scatterplots of individual DFA values as a function of GS corroborated the notion of an overall positive correlation across frequencies as well as a salient quadratic component in MEG at 10 Hz (Fig. 2e, f). To further confirm these observations, we also computed correlations between GS and DFA values within the canonical frequency bands established in a data-driven manner with Louvain clustering (Supplementary Fig. 1) and observed significant linear positive correlations in all frequencies in MEG and in theta (θ, 4–8 Hz), alpha (8–14 Hz) and beta (β, 15–29 Hz) in SEEG (Supplementary Fig. 2). We further confirmed that results did not depend on the choice of the phase synchronization metric by also estimating GS with PLV for MEG and with wPLI for SEEG (Supplementary Fig. 3). Moreover, we also tested for the generalization of these findings into time-lagged forms of neuronal synchronization. We computed lagged synchronization for lags of 1 and 2 cycles (See Methods) and observed GS of lagged synchronization both in MEG and SEEG peaking in the alpha band (Supplementary Fig. 5a, c). The linear correlations between DFA and lagged synchronization were very similar to those observed for non-lagged synchronizations (Supplementary Fig. 5b, d).

As both synchronization and LRTCs have been shown to be trait-like phenomena, we further investigated whether their correlations would also exhibit the high test-retest reliability that characterizes trait-like phenomena. Using the Gauge Repeatability method[46], we confirmed that both the individual GS and DFA values (Supplementary Fig. 4a–c) and their correlations (Supplementary Fig. 4d, e) had significant test-retest reliability and capacity. In the light of the modeling results, the positive linear correlations between GS and mean DFA thus strongly suggest that healthy brain networks operate mostly on the subcritical side of an extended critical regime, or GP, while the quadratic correlations in MEG indicate that some subjects also operate around the peak of this regime. The robust co-variability between the strengths of neuronal synchronization and LRTCs implies that there is a large range of possible individual operating points. This supports our hypothesis that the human brains operate in an extended critical regime, such as the Griffiths Phase, instead of being confined to the vicinity of a singular critical point.

### Synchronization and LRTCs are positively correlated across brain regions

To get insight into the brain anatomy and localization of these correlations, we estimated the correlation of narrow-band oscillation synchronization and DFA exponents across subjects separately for each parcel by estimating a mean nodal synchronization using Node Strength (NS; with MEG parcels or SEEG contacts being the nodes) and its correlations with the DFA exponents of the node within the canonical frequency bands. Positive linear correlations between NS and DFA exponents were significant in all frequencies except in the delta-band and within all functional subsystems in MEG data yielding on average Pearson correlations of 0.5 (Fig. 3a). Significant linear correlations were observed in parcels throughout the cortex. In the alpha band, correlations were the strongest in dorsal visual areas and prefrontal cortex (PFC), and in the high-gamma-band in left-parietal and posterior regions (Fig. 3b). In the beta band, these correlations were strongest in the somatomotor (SM) regions, and in the gamma-band, in the ventral visual stream regions, PFC, and cingulate structures (Supplementary Fig. 6a). In SEEG, positive linear correlations were observed in the delta to beta frequency bands, and in ripple-high-gamma frequencies (Fig. 3c). Positive linear correlations in theta and alpha bands were strongest in medial prefrontal, temporal, and parietal regions belonging to the default mode (DM) network (Fig. 3d and Supplementary Fig. 6b).

Negative quadratic correlations indicative of operation close to the critical point were observed from alpha to gamma bands in MEG (Fig. 3e) and were strongest in posterior regions in alpha and in temporal and medial regions in the high-gamma-band (Fig. 3f and Supplementary Fig. 6c). Significant quadratic correlations in SEEG were much sparser (Fig. 3g) and observed in parcels mostly in parietal and frontal regions (Fig. 3h and Supplementary Fig. 6d). These results were reproduced also using individual wavelet frequencies (Supplementary Fig. 7) and had significant test-retest reliability (Supplementary Fig. 4f–i). The spatiotemporally widespread co-variability between synchronization and LRTCs thus further supports the idea that also large areas of the human neocortex are more likely to operate in a Griffiths-phase critical regime instead of at a fixed critical point. The anatomical heterogeneity and frequency-specificity of these anatomical patterns also support the hypothesis that different brain structures or functional systems may express distinct and partially independent operating points[16].

### Negative correlations between synchronization and LRTCs characterize the Epileptogenic Zone (EZ)

Epilepsy has been associated with excessive excitation, hypersynchronization[47–49], and altered DFA exponents[31,50]. Hence, we hypothesized that brain areas in the epileptogenic zone (EZ) could be characterized by operating points in the supercritical side of the critical regime, in contrast with the healthy brain areas that appear to operate mostly in the subcritical side (Fig. 4a). We thus investigated correlations between synchronization and LRTCs in SEEG contacts in EZ and compared these correlations with those for non-EZ contacts (Fig. 4b). The EZ contacts exhibited slightly larger PLV values but similar DFA exponents compared to the non-EZ contacts (Fig. 4c, d). Importantly, as hypothesized, the correlations between subjects' GS and mean DFA were negative for EZ contacts in most frequencies (Fig. 4e). The difference in mean correlation values between non-EZ and EZ was significant between 3 and 8 Hz (peaking at 4 Hz with $p_{perm} < 10^{-7}$) and at 165 Hz ($p_{perm} < 0.05$) (Fig. 4f).

## Discussion

In the human brain, there is considerable variability among healthy individuals in the strengths of long-range phase synchronization and LRTCs[10–16]. Both synchronization and LRTCs are test-retest reliable[12,51,52], heritable[12,53–55], and influenced by genetic polymorphisms[15], indicating that this variability is trait-like and rooted in the individual functional neuroanatomy rather than attributable to moment-to-moment variability or measurement noise. Discovering the neuronal dynamics basis defining the boundaries of such variability is essential both for understanding the neurobiological factors underlying individual differences in cognitive performance and for developing new therapeutic approaches for brain diseases characterized by abnormal synchronization levels. We tested here the hypothesis that individual variability in observable inter-areal synchronization and oscillation LRTCs in vivo would be explained by the individual's position in the critical regime of the state space (Fig. 1a, b), i.e., by the individual operating point, which is determined by the underlying control parameters. Moreover, we hypothesized that the structural and functional heterogeneities in human brains would lead to the stretching of a theoretical, critical point into a regime of critical-like dynamics, the Griffiths Phase (GP), which would enable the expression of individual operating points along a wider regime. This has previously been suggested by modeling studies, where a GP has been shown to arise, e.g., from a hierarchical modular organization of the structural connectome[37–40]. If the variability of synchronization levels would be explained by the variability in the individual operating points, inter-areal oscillatory synchronization levels should be correlated with LRTCs across both individuals and brain regions.

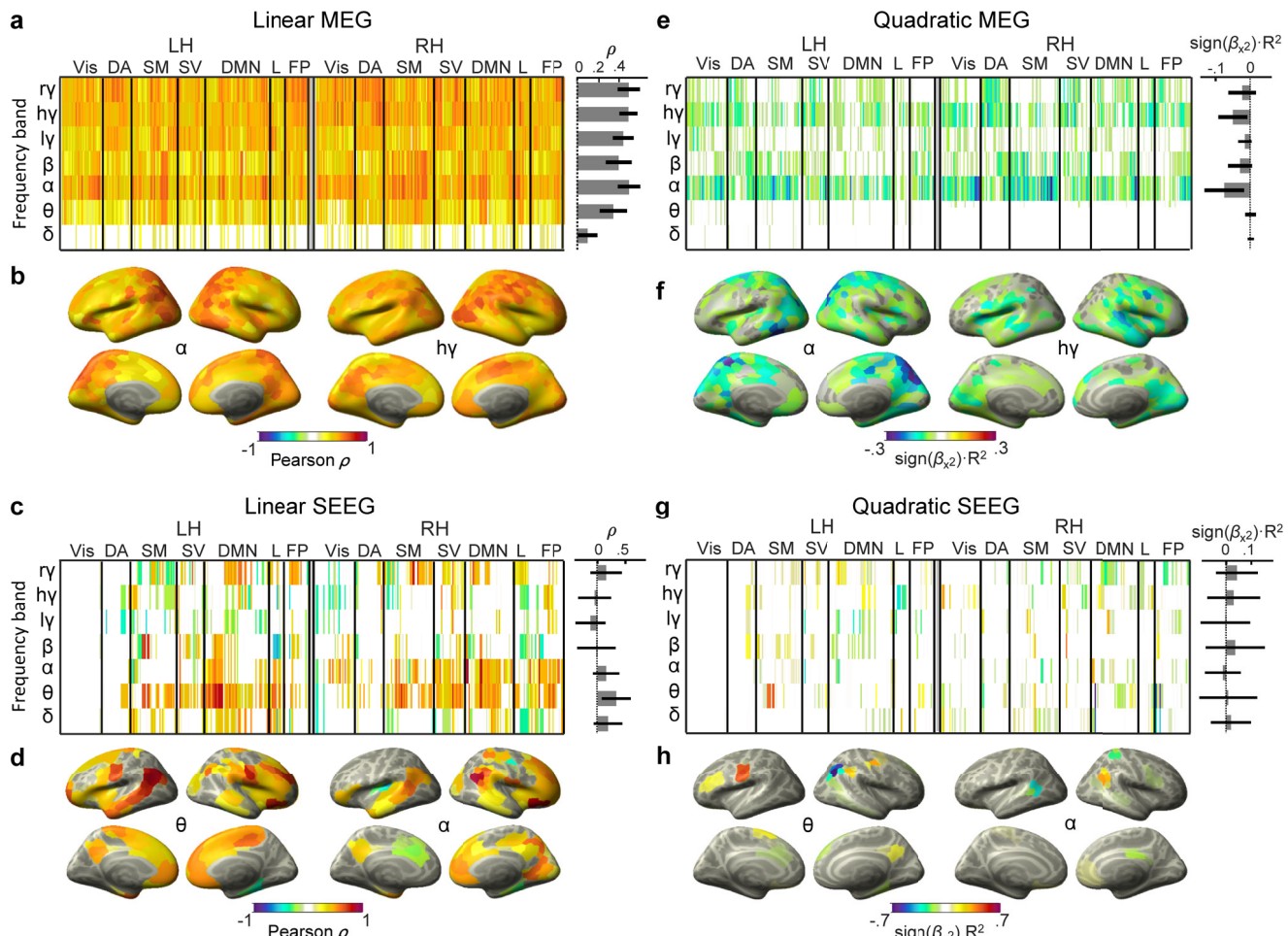

**Fig. 3 | Correlations of synchronization and LRTCs at parcel-level. a** Linear correlation (Pearson' $\rho$) of node strength (NS) with mean DFA exponents estimated for each parcel in MEG data ($N = 192$ recordings) for left and right hemispheres (LH and RH) averaged over canonical frequency bands. Non-significant correlations are masked in gray. The gray bars on the right show the mean correlation ± SD across significant parcels ($N = 400$ parcels) for each band. **b** Cortical topographies of the correlation across datasets between NS and DFA in MEG for alpha and high-gamma frequency bands; shown only for parcels where correlation is significant. **c, d** Same as in **a, b**, for theta and alpha bands in SEEG ($N = 57$ recordings). **e–h** Same as above for partial-quadratic correlations. Frequency bands: δ: 2–4 Hz; θ: 4–8 Hz; α: 8–14 Hz; β: 15–29 Hz; lγ: 30–70 Hz; hγ: 71–135 Hz; rγ: 165–225 Hz. Functional networks: Vis visual, DA dorsal attention, SM somatomotor, SV saliency/ventral attention, DMN default mode network, L limbic, FP frontoparietal/control. Source data for panels **a, c, e, f** is provided in the source data file.

We first showed, using computational modeling of macroscopic oscillation synchronization dynamics, how the operating point within a GP determines distinct modes of covariation of long-range phase synchronization and LRTCs in local oscillation amplitude fluctuations. We then leveraged the large interindividual variability across subjects in the LRCTs and synchronization levels to investigate whether they co-vary within individuals and regions in MEG and SEEG data and thereby deduced their in vivo operating regimes. We found that in healthy subjects and in healthy brain areas of epileptic patients, inter-areal synchronization of neuronal oscillations was positively correlated with LRTCs in all frequencies above 5 Hz in MEG and in the alpha and gamma bands in SEEG (see Fig. 3). In addition, we also observed a significant quadratic trend, which indicates that a subset of subjects operates around the peak of the critical regime, exhibiting the strongest LRTCs and thereby operating closest to the theoretically attainable criticality (see Fig. 1c). These findings strongly suggest that individual and regional variability in the synchronization levels is explained by the variability in the operating points in line with the GP framework.

The GP framework extends the classical notion of criticality being constrained to a singular point in the space of control parameters which underlies hypotheses of the brain operating either at the critical point[34], near the critical point[17], or in a slightly subcritical regime[41,56,57]. Importantly, both here and in our earlier study[16], we found heterogeneity in dynamics as well as covariance of LRCTs and synchronization levels also across neocortical regions within specific frequency bands. This suggests that the different functional brain systems in individual subjects operate in different positions of the critical regime[16] rather than being fully controlled by a global parameter. These findings are consistent with a computational study showing with the Landau-Ginzburg model that phase transitions may not be transitions between quiescent and fully activity states, but rather transitions of synchronization emerging from the noninfinite size of mesoscopic regions and spatial dependence[44]. A simulation study using detailed microcircuit data has demonstrated that at a sharp transition from synchronous to asynchronous activity, a spectrum of network states emerges due to a range of neurophysiological mechanisms[58]. At the large-scale systems level, tuning a computation model close to a supercritical Hopf bifurcation maximized its fit to BOLD signal functional connectivity[59], implying that brains indeed operate near criticality.

The present results converge to show that healthy brain areas operate mostly in the subcritical side of the GP, which is in line with previous studies emphasizing the possibility and importance of

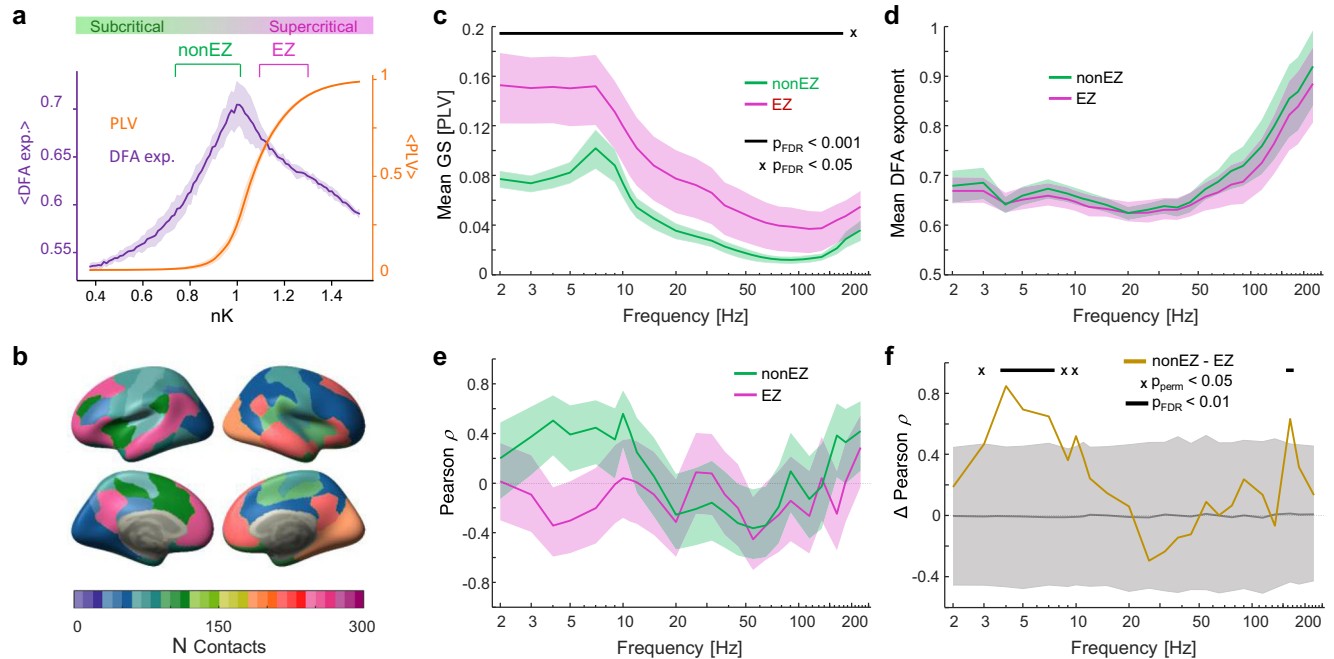

**Fig. 4 | Correlations of synchronization and LRTCs for epileptic zone SEEG contacts. a** Schematic illustrating our hypothesis of the healthy non-EZ areas operating mostly in the subcritical side of the GP and epileptic zone contacts (EZ) areas operating on the supercritical side. Model data as in Fig. 1c. Mean PLV and DFA are presented with 90% CI. **b** Cortical distribution of epileptic zone (EZ) contacts across functional networks. **c** Mean graph strength (GS) estimated with PLV and **d** mean DFA scaling exponents for non-EZ (orange) and EZ (purple) contacts (with 95% confidence intervals). The black line and "x" at the top represent significant group differences at $p_{FDR} < 10^{-3}$ and $p_{FDR} < 0.05$, respectively (case-resampling permutation test, two-sided, significant after FDR correction with Benjamini–Hochberg). **e** Mean linear correlation coefficients (Pearson's $\rho$, with 95% confidence intervals) between GS and DFA exponents for non-EZ and EZ. **f** Difference in correlation coefficients between non-EZ and EZ, with a gray area representing percentiles 2.5 to 97.5 of a surrogate difference distribution obtained with case-resampling permutations. The black line at the top indicates $p_{FDR} < 0.01$ (same test as in **c, d**), "x" indicates $p_{perm} < 0.05$. Source data for panels **c**–**f** is provided in the source data file.

healthy mammalian brains to operate in a subcritical-like regime. Prior modeling and in vivo studies from normal human and non-human brains have suggested this operating point to be in the subcritical phase very near the critical point[41,56,60]. A subset of the healthy subjects contributed also to a quadratic relationship between synchronization and LRTCs similar to prior observations of quadratic correlation between oscillation amplitudes and LRTCs in human MEG and EEG[61], in intracranial EEG[49], and in in vitro responses to the manipulation of the E/I balance[62,63]. Our findings also extend previous findings showing covariation of evoked activity and synchronization in electrocorticography[49], and systematic covariation in the decay of spatial and temporal correlations in the human cortex that suggested that this covariation is linked to intrinsic excitability measures tracking antiepileptic drug action[36]. However, here we posit that a singular operating point in the subcritical phase would be biologically sub-optimal because it would be associated with attenuated propagation of neuronal processing, inadequate synchronization, and loss of power-law scaling that endows the functional benefits of criticality. Operation in the subcritical-to-critical side of a GP, on the other hand, yields a range of operating points that are sufficiently far from the supercritical regime that may predispose to epileptogenesis[49,64] while benefitting from the functional advantages that have been ascribed to operation at the classical critical point, such as maximal dynamic range, maximal capacity for information transmission and storage, and optimal computational efficiency[2,17,29,62]. While the GP has already been established to exhibit critical-like dynamics, its functional implications have not been fully explored. We propose here that the extended critical regime, i.e., the GP in human brain activity, would have functional benefits akin to those proposed for the classical critical point that are necessary for healthy brain functioning. Thus, a fundamental functional implication for operation in the GP is that it would enable

the shifting of the individual or even brain-system-specific operating point dynamically in response to endo- or exogenous demands without losing the functional advantages conferred by critical-like dynamics.

We observed variability in the strength of correlations not only across brain regions but also across frequencies, indicating that brain functional networks have a small variability in their operating points in line with the GP framework where the GP arises due to functional and structural heterogeneity[37]. The correlations were linearly positive at the whole-brain level and in most parcels and frequencies. However, in alpha, beta, and high-gamma bands there was also a quadratic trend, showing that in a subset of subjects, these functional networks operated around the peak of the extended critical regime. Given the fundamental functional roles that alpha and gamma-band oscillations are thought to play in feed-back and feed-forward information processing[65–69], respectively, it is conceivable that these oscillations operate around the peak of the putative critical regime, enabling maximal dynamic range and flexibility. The localization of strongest alpha-band correlations into dorsal visual areas and PFC and gamma-band correlations into the ventral stream areas is in line with previous findings of the differential localization of power and synchronization in these frequencies[5,70–72] and with their biologically differential dependence on anatomy, neurobiological mechanisms and neuro-modulatory genes[15,73,74]. These findings of heterogeneous neuronal dynamics are also in agreement with a previous study from awake mouse visual cortex where scale-free neural activity was limited to specific subsets of neurons[75]. Heterogeneity across frequencies and functional neuroanatomy could provide a means for adapting behavior according to environmental demands.

In contrast to healthy brain activity, epilepsy has been associated with a shift in the excitation-inhibition balance towards excessive

excitation, which leads to aberrant pathological brain dynamics[47,48,76] and episodes of abnormal hyper-synchronous activity[64]. Brain criticality is primarily thought to be controlled by the finely balanced E/I ratio and excessive excitation may lead to operation in the super-critical phase, as has been shown in in vitro (Plenz & Thiagarajan[25]; Toker et al.[41]; Yang et al.[77]) and modeling (Poil et al.[31]) studies. We thus investigated whether inter-ictal brain dynamics in epileptogenic brain regions would be characterized by supercritical-like dynamics. This would be indicated by a negative relationship between synchronization and LRTCs (see Fig. 1c). We found that the EZ contacts were indeed associated with a negative correlation between synchronization metrics and the DFA scaling exponents (LRTCs). This constitutes empirical evidence for epilepsy being associated with the epileptogenic circuitry operating in the supercritical side of an extended critical regime, which may be a key factor that predisposes these circuits to generate epileptic seizures. In SEEG, in the nominally healthy non-EZ contacts outside of the epileptogenic zone, we found no evidence for quadratic correlations that would indicate operation near the peak of the critical regime. Antiepileptic drugs, used also by the patients in this study, have been shown to push the brain towards subcriticality[36,78] and may thus constitute a likely explanation for this phenomenon. Another possible contributing factor might be long-term neuronal plasticity that may have globally enhanced inhibitory connections in order to compensate for locally increased excitability in the epileptogenic network, which would drive these healthy brain regions towards the subcritical side of the critical regime. This is supported by the presence of negative correlations as indicative of supercritical dynamics in gamma-band in some brain areas, suggesting an imbalance in the system dynamics.

Given that brain criticality is primarily thought to be controlled by the E/I ratio, where an imbalance of E/I or synchronization leads to sub- or supercritical dynamics[25,31,63,79] and given that pathological human brain activity is associated with changes in brain E/I balance[9,80,81], we propose that pathological synchronization dynamics (hypo- or hyper-synchronization) in brain diseases[9,50,61,64,76,80,81] could emerge via modulations of brain critical dynamics by changes in the E/I balance or by other physiological parameters such as changes in structural connectivity.

In conclusion, we find that synchronization levels and LRTCs are correlated across subjects in a manner which suggests that healthy human brains operate in the subcritical side and around the peak of an extended critical regime, the Griffiths Phase (GP), while epileptogenic areas operate in the supercritical side. These findings provide strong evidence that the variability in synchronization levels is determined by an individual's operating point, i.e., by the individual position in the GP.

## Methods
### Overview
We first used a computational model based on a hierarchical Kuramot-based computational model to generate brain dynamics in a Griffiths Phase (GP) and in the classical criticality framework and to investigate the relationship between oscillatory inter-areal synchronization and LRTCs across wide range operating points near the critical point. We then estimated inter-areal synchronization and LRTCs in source-reconstructed resting-state magnetoencephalography (MEG) data recorded from healthy subjects and resting-state stereo-electroencephalographic (SEEG) data recorded from epilepsy patients. Simulated, MEG, and SEEG data were filtered into narrow-band frequency time series. Synchronization was computed between all pairs of brain regions/electrodes and LRTCs were analysed with detrended fluctuation analysis (DFA) for each parcel. To test whether the heterogeneity of the individual variability of LRCTs and synchronization levels reflects the variability of individual operating points as predicted by the GP framework, we computed linear and quadratic correlations between synchronization and

LRTC metrics. In SEEG data, we carried these analyses out separately for EZ and non-EZ networks and computed the difference in correlation coefficients.

### Modeling brain dynamics and criticality with a hierarchical Kuramoto model
A hierarchical Kuramoto model was used to simulate coupled neuronal population narrow-band oscillation dynamics with LRTCs and to investigate observable correlations between synchronization and DFA exponents. The Kuramoto model is a parsimonious dynamical model for neural oscillation and synchronization. Despite its simplicity, the Kuramoto model is capable of, like many other biophysically detailed models, capturing rich neuronal dynamics, including mesoscopic[43] and macroscopic brain waves, with a wide array of bifurcation types that are analytically tractable. Importantly, the Kuramoto model was recently shown to produce GP-like dynamics in large synthetic hierarchical networks[38,82,83]. Therefore, the Kuramoto model is a relevant and potent dynamical model when combined with real human structural connectome to address the GP.

We adapted a two-layer nested Kuramoto model of local and large-scale neuronal oscillatory dynamics with a hierarchical variant[15,42] consisting of 400 nodes (corresponding to 400 brain areas), each containing a conventional Kuramoto population of oscillators, $N = 500$. Oscillators were modeled with a Gaussian frequency distribution with a mean value of 10 and a standard deviation of 3 to obtain properties of an oscillating network with coupled interactions, where the oscillation frequency per se does not play a role. The phases of all oscillators in a node were averaged to derive a single time series per region, whose absolute values were then taken to obtain the Kuramoto order parameter, corresponding to the amplitude envelope of local oscillations. In this model, a local control parameter K determines the homogeneous coupling strength within the nodes while a global control parameter L determines the inter-regional coupling strength based on the pairwise-connectivity white-matter axonal fiber counts estimated with structural diffusion tensor imaging (DTI) and obtained from the Human Connectome Project (HCP). White noise was added to the final time series to simulate device and environmental noise. This model yields dynamics both at local and large-scale network levels that are comparable to the empirically observable dynamics within and between cortical regions. Thus, we were able to model both local interactions within smaller regions (corresponding to cortical MEG parcels or SEEG contacts), and synchronization (corresponding to that between parcels or contacts across the whole brain). We assessed LRTCs for each region using DFA, and synchronization between all pairs of regions using the PLV (see below).

### Diffusion tensor imaging (DTI) data
Briefly, we computed structural connectomes for 57 unrelated subjects randomly selected from the WU-Minn dataset (1200 subjects) of the Human Connectome Project (HCP, https://www.humanconnectome.org). From each subject's minimal-preprocessed Diffusion Weighted Imaging (DWI) data, we generated a preliminary tractogram of 50 million streamlines using Multi-Shell Multi-tissue (MSMT) spherical deconvolution and probabilistic tractography (maximum tract length = 250 mm; fiber orientation distribution (FOD) amplitude cut-off, 0.01; seeding from the gray matter-white-matter interface; application of the anatomically constrained tractography (ACT) framework[84]. Using spherical-deconvolution informed filtering (SIFT), we filtered the initial tractogram in order to reduce bias derived from longer, thicker tracts. Finally, we created individual cortical parcellations (with 400 parcels) based on the Schaefer atlas and then collapsed streamlines to weighted edges between parcels. Each end of a streamline was assigned to the most probable parcel using a 3 mm radial search outwards, and the weight of the edge between each pair of parcels was set as the number of streamlines connecting them.

Weights on the matrix diagonal (representing self-connections) were set to zero, so that in simulations, each parcel (node) is connected to all others, but without local feedback.

## Acquisition of MEG and MRI Data

We recorded MEG data from 52 healthy subjects (age: 31 ± 9.2, 27 male) during a 10-min eyes-open resting-state session with a Vectorview/Triux (Elekta-Neuromag/MEGIN, Helsinki, Finland) 306-channel system (204 planar gradiometers and 102 magnetometers) at the Bio-Mag Laboratory, HUS Medical Imaging Center, Helsinki. Overall, 192 sessions of MEG data were obtained, with subjects contributing, on average, 3.7 ± 4 sessions each. Subjects were instructed to focus on a cross on the center of the screen Bipolar horizontal and vertical electrooculography (EOG) were recorded for the detection of ocular artifacts. MEG and EOG were recorded at a 1 kHz sampling rate. For each subject, T1-weighted anatomical MRI scans (MP-RAGE) at a resolution of $1 \times 1 \times 1$ mm with a 1.5-Tesla MRI scanner (Siemens, Munich, Germany) were obtained at Helsinki University Central Hospital for head models and cortical surface reconstruction. The study protocol for MEG and MRI data was approved by the Coordinating Ethical Committee of Helsinki University Central Hospital (ID 290/13/03/2013), written informed consent was obtained from each subject prior to the experiment, and all research was carried out according to the Declaration of Helsinki.

## MEG data preprocessing and source modeling

Temporal signal space separation (tSSS) in the Maxfilter software (Elekta-Neuromag) was used to suppress extracranial noise from MEG sensors and to interpolate bad channels. Independent components analysis (ICA) adapted from the Fieldtrip toolbox) was used to extract and identify components that were correlated with ocular artifacts (ocular, heartbeat, or muscle artifacts). Volumetric segmentation of MRI data, flattening, cortical parcellation, and neuroanatomical labeling with the 400-parcel Schaefer atlas was carried out with the FreeSurfer software. The MNE software was then used to create cortically constrained source models with 5-mm inter-dipole separation, for MEG–MRI colocalization, and for the preparation of the forward and inverse operators. We computed noise covariance matrices (NCMs) using the preprocessed MEG data filtered with finite-impulse-response (FIR) filters at 151–249 Hz, averaged across 10 s time windows. NCMs were used for creating one inverse operator per session and the dSPM method with regularization parameter $\lambda = 0.11$. We then estimated *vertex fidelity* to obtain fidelity-weighted inverse operators that reduce the effects of spurious connections resulting from source leakage and collapsed the inverse-transformed source time series into parcel time series in a manner that maximizes the source-reconstruction accuracy[42,85]. For each parcel pair (edge) we also computed *cross-parcel phase-locking* of the reconstructed simulated time series, reflecting cross-parcel signal mixing, and excluded parcels and edges with low fidelity and high cross-parcel phase-locking, using individual thresholds to retain for each subject the top 90% parcels by fidelity and the bottom 95% of edges by cross-parcel mixing (14.9 ± 0.2% of parcels and 14.1 ± 0.1% of edges rejected on average per set).

## Acquisition of SEEG data

We recorded 10 min of stereo-EEG (SEEG) neuronal signals from 68 drug-resistant focal epileptic patients (age: 30 ± 9.4, 38 male) during the clinical assessment of the epileptogenic zone (EZ) at the "Claudio Munari" Epilepsy Surgery Centre in the Niguarda Ca' Granda Hospital, Milan. These resting-state recordings were free of seizure activity, and there were no seizures within one hour prior to or after the recording. During the recording, the patient was asked to lay down in their bed in a quiet resting state with their eyes closed.

Visual inspection of delta amplitude profiles from EEG channel pair C3-P3 and video recordings of the patient were used by trained technicians to ascertain the absence of signs of sleep or drowsiness. Intracranial monopolar (with contacts sharing the reference to a single white-matter contact) local-field potentials were acquired from brain tissue with platinum–iridium multi-lead electrodes. Between 8 to 15 contacts, each 2 mm long, 0.8 mm thick, and with an inter-contact border-to-border distance of 1.5 mm (DIXI medical, Besancon, France), were present in each penetrating shaft, with the amounts of electrodes and their anatomical positions varying according to surgical requirements[86]. Each subject had 17 ± 3 (range 9–23) shafts with a total of 153 ± 20 electrode contacts on average. The electrode positions were localized after implantation using CT scans and the SEEGA automatic contact localization. Structural MRIs were recorded before implantation and colocalized with post-implant CT scans using rigid-body coregistration[87]. Individual patients' contacts were assigned to parcels of the Schaefer atlas.

We acquired an average of 10 min of uninterrupted spontaneous resting-state activity with eyes closed with a 192-channel SEEG amplifier system (Nihon-Kohden Neurofax EEG-1100) at a sampling rate of 1 kHz. All patients were taking antiseizure medications (antiepileptic drugs (AEDs) with a large variation in the dosage and compounds (See Supplementary Table 1 for the dosage administered in the morning of the day of the recording), and the time elapsed the last drug administration and the SEEG resting-state recording was not controlled. Patients gave written informed consent for participation in research studies and for publication of results pertaining to their data. The ethical committee of the Niguarda Hospital, Milan, approved this study (ID 939) which was performed according to the Declaration of Helsinki.

## Filtering, preprocessing, and identifying and analysing the epileptogenic zone (EZ)

The EZ is clinically defined as the brain regions where ictal activity initiates and propagates[88]. In this work, the EZ in individual patient brains were identified and stringently confirmed by clinicians using peri-ictal and ictal SEEG recordings[89]. Defective contacts that demonstrated non-physiological activity (1.3 ± 1.2, range 0–10) were excluded from analyses, and three subjects in which more than 50% of contacts were defective were discarded from further analyses. Analyses in Figs. 2, 3 included only contacts from tentatively healthy regions (nEZ), i.e., those in which ictal activity was not observed during SEEG monitoring, while analysis in Fig. 4 were performed for EZ contacts. Subjects who had more than half of their contacts in the EZ (eight subjects) were excluded from the analysis of non-EZ data and included in EZ data. For non-EZ data, we included data from 57 subjects, with a total of 4453 non-EZ contacts (average per subject 78 ± 19, range 41–124). For EZ, we used only the contacts defined as being epileptogenic– i.e., contacts that were located within the EZ or were part of the seizure propagation network and all contacts from eight subjects in whom >50% of contacts had previously been classified as EZ. Subjects who had <11 EZ contacts were excluded. Thus, both analyses had 57 subjects, with 49 common to both non-EZ and EZ analyses, eight included only in non-EZ, and eight only included in EZ analysis. The total number of EZ contacts was 1725 (average per subject 30 ± 17.5, range 11–79). The average distance between EZ and non-EZ contacts was similar (Supplementary Fig. 8).

As occasional inter-ictal events characterized as large amplitude spikes or sharp waves with wide spectral and spatial spread may bias the LRTC and phase synchronization estimates[64], we rejected temporal segments with such activity[45]. Briefly, we partitioned the signals into adjacent time windows of 500 ms and decomposed the signals into 24 frequency bands with Morlet-wavelet filtering. With this time-frequency decomposition, we first computed the mean and the standard deviation of the signal amplitude for each individual SEEG gray matter electrode contact and frequency. A given 500 ms time window was considered putatively contaminated by epileptic (high-amplitude and spike-like) artifacts and rejected if ≥10% of contacts exhibited an

amplitude greater than their mean amplitude plus five times the standard deviation, and this effect was found in ≥25% of the frequency bins. We then referenced SEEG electrodes in gray matter to the closest contacts in white matter, which yields signals with more accurate phase estimates[47], FIR-filtered broad-band contact time series with a cutoff at 440 Hz and removed 50 Hz line noise and its harmonics with notch FIR filters[45].

## Analysis of inter-areal synchronization

MEG and SEEG data were filtered into complex-valued narrow-band time series using 24 Morlet wavelets (parameter $m = 5$) with logarithmically increasing center-frequencies between 2–225 Hz. We computed pairwise phase synchronization for all narrow-band frequencies between all 400 cortical parcels in source-reconstructed MEG data and between all non-epileptogenic contact-pairs of SEEG data for all narrow-band frequencies. The main analysis was based on phase synchronization estimated with the phase-locking value (PLV) for SEEG data and with the weighted Phase-Lag Index (wPLI) for MEG data, but for comparison, phase synchronization using the PLV for MEG and the wPLI for SEEG are shown in Supplementary material. The wPLI, unlike PLV, discards zero-lag coupling that in MEG data is mostly spurious caused by linear mixing, but still can detect true positive coupling when there is a small phase lag[90]. We also investigated delayed synchronization by computing the same synchronization metrics where one time series had been shifted towards the other by a lag of either one or two cycles (Supplementary Fig. 5). To assess the whole-brain and nodal level of synchronization, we used graph theory. Node Strength $NS$ was obtained for each node (contacts in SEEG and parcel in MEG) by averaging the strength of edges (connections of synchronization) for that node. For each subject, all the cortical NS values were then averaged again to estimate the Graph Strength $GS$ for each frequency that defines the whole-brain level of synchronization,

## Detrended fluctuation analysis

We used Detrended Fluctuation Analysis (DFA) to estimate monofractal scaling exponents of neuronal LRTCs that typically vary between 0.5 and 1[30,52,55,64]. DFA was carried out in the Fourier domain[91] with a Gaussian weight function used for detrending and using 25 loglinear windows from 5 to 56 s. The fluctuations were fitted with a robust linear regression with a bisquare weight function to obtain the DFA exponents, all with negligible fit error. DFA exponents were computed for all contacts of all SEEG subjects and all parcels of all MEG sets, and for each narrow-band frequency. Mean DFA scaling exponents were obtained by averaging DFA exponents over all nodes. We then removed outliers >3 SD from the median. As inter-day variability was non-neglectable for violations of independence[51] (Supplementary Fig. 4), multiple recording sessions from the same MEG subjects were treated as individual data points.

## Correlation of synchronization and LRTCs

Pearson's linear correlation analysis was used to estimate the correlation between subjects'/sets' GS and mean DFA values for each frequency. To obtain the surrogate distribution of correlation coefficients, the order of the dependent variables was shuffled 1000 times. Multiple hypothesis testing was corrected with the Benjamini–Hochberg method, pooling together both synchronization metrics and all frequencies. We also estimated correlations between GS and mean DFA with a partial-quadratic model, i.e., a purely quadratic correlation where the linear trend in the dependent variables was partialed-out. To test whether the relationship between synchronization and criticality was concave and not convex (peaks rather than dipping, with the concave inverted-U curve opening down), in addition to obtaining the $R^2$ statistic, the coefficient of the quadratic term was multiplied with the sign (note that the y-axis is reversed in the main

text subject correlations figure panels, so that the negative values indicating concave correlations are on top).

We next calculated linear and partial-quadratic correlations across subjects for each cortical parcel and frequency. In MEG data, for each of the 400 parcels, its local $NS$ value was correlated with its DFA exponent across sets, with outlier rejection, surrogate calculation, and FDR correction (this time including also the 400 parcels), same as above. In SEEG data, contacts from all subjects were pooled into parcels of the 100-parcel Schaefer atlas, and the correlations between parcel $NS$ values and DFA exponents were computed for all parcels containing at least 5 electrodes (after outlier rejection, resulting in 77/100 parcels). In order to be able to compare results between MEG and SEEG, SEEG data were interpolated into the atlas of 400 parcels that was also used for MEG. Linear and partial-quadratic correlations were then computed for these parcels in the same way as described for the subject level. For visualization purposes, we grouped frequencies in data-driven bands, individuated as the optimal community structure determined by the Louvain method of the self-similarity frequency-by-frequency matrix of the linear parcel correlations (delta, δ: 2–4 Hz; theta, θ: 4–8 Hz; alpha, α: 8–14 Hz; beta, β: 15–29 Hz; low-gamma, lγ 30–70 Hz; high-gamma, hγ: 71–135 Hz; ripples-gamma, rγ: 165–225 Hz). Results were similar for single frequencies not grouped into bands (Supplementary Fig. 7) and if statistics were averaged into bands after correlations instead of before.

## Reporting summary

Further information on research design is available in the Nature Portfolio Reporting Summary linked to this article.

## Data availability

Raw electrophysiological data cannot be shared publicly due to regulations imposed by the Ethical Committees but can be shared for collaborative efforts upon request.

A minimal dataset that can be used to reproduce the main findings of this study, containing phase synchronization matrices and DFA exponents for MEG and SEEG cohorts along with supporting data, as well as simulated data, is publicly available at DataDryad repository (https://doi.org/10.5061/dryad.vdncjsxzn).

Source Data for the main manuscript figures are provided as a Source Data file. Source data are provided with this paper.

## Code availability

All code used in this work to produce the modeling results, connectome matrices, to final figures can be found at https://github.com/palvalab/DFA_Synch/.

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

## Acknowledgements

This work was supported by grants from the Academy of Finland (SA 1266745, 1296304 to J.M.P. and SA 325404 to S.P.), from the Finnish Cultural Foundation to S.H.W. (postdoc fellowship 00220071), and from the Sigrid Jusélius Foundation to S.P. and J.M.P.

## Author contributions

J.M.P. and S.P. conceived the study. F.S., G.A., and L.N. procured data. F.S., G.A., and S.H.W. performed data preprocessing. M.F. performed data analysis. V.M. performed computational modeling. M.F., F.S., S.H.W., J.M.P., and S.P. wrote the manuscript. All authors approved the final manuscript.

## Competing interests

The authors declare no competing interests.
