## [Peer Review File · Nature Communications]

Brain criticality predicts individual levels of inter-areal synchronization in human electrophysiological dataReviewer #1 (Remarks to the Author):

The manuscript provides a report on the relationship between synchronization levels and Hurst exponents in MEG and SEEG. The authors motivate their study by observations from a network model consisting of Kuramoto oscillators which exhibits the well-known synchrony phase transition and a peak in the Hurst exponent at criticality. The authors argue that these two metrics may consequently be used in empirical data to determine where dynamics resides in phase space, specifically with regards to criticality. The authors report mainly linear positive relationships as well as quadratic relationships between synchrony and Hurst exponent, thus suggesting that networks reside in the subcritical or close to critical regime.

I appreciate the authors' effort to provide a comprehensive study of these two metrics (synchrony, Hurst exponent) across subjects, brain regions and methods (MEG, SEEG). Although one would ideally have the means to tune or perturb brain networks directly and within a subject in order to determine whether there are coherent changes in the two metrics, the present study provides a comprehensive effort to investigate these dynamics across subjects. It assumes that there is a regime where both metrics change in predictable ways.

- There seem to be some frequency bands that behave opposite to what is reported as the main finding. For example, there are negative correlations present in low and high gamma bands in SEEG. How can this be explained and reconciled under the present framework? Is the definition of criticality and where it occurs potentially frequency dependent? Please elaborate more on this as it is central to your interpretation of the findings.

- Negative quadratic correlations indicative of close proximity to a critical point seem to be more present in MEG, less so in SEEG (e.g. Fig. 2 E-H, Fig. 3 E-F), if I understand the authors correctly. An important difference between MEG (which here is performed on healthy subjects) and SEEG (which is performed on epilepsy patients) is that patients with SEEG are consequently mostly under antiepileptic drug (AED) treatment. AEDs have been shown to effect LRTCs, long-range spatial correlations and synchrony in predictable ways, i.e. suggestive of a shift towards subcritical dynamics (e.g. <https://doi.org/10.1073/pnas.1911461117>).

Have the authors taken AED into account in their analyses? Is it possible that the more linear (and less quadratic) relationship observed in SEEG may similarly indicate a more subcritical state, potentially driven by AEDs?

- The authors discuss Griffith spaces and see their results as support. Specifically, they state: "To the best of our knowledge, these findings thus constitute the first empirical evidence towards the discovery of Griffiths-phase-like dynamics in the brains in vivo." I am not sure how the data provided may prove or disprove these as the present study mainly demonstrates covariation of different types. It has been shown that dynamics, including synchronization levels and related criticality measures, fluctuate widely across patients, regions and time, e.g.

<https://doi.org/10.1073/pnas.1513716112>, Fig. 3. Could the authors please elaborate their point more and how it differs from these earlier findings and/or transcends them?

- The effect of interictal activity: The authors state that channels with interictal discharges were removed from the analysis. How was this done for the channels in the EZ, as the EZ by definition contains most of these discharges? How did this removal impact the results?

- Were synchrony levels overall higher (and Hurst exponent levels overall lower) in EZ as would be assumed by this picture or framework that authors propose? If not, how can this be reconciled?

- Synchrony levels may depend on the spatial sampling, i.e., be higher when electrodes are placed more closely to each other. In SEEG, it is often attempted to capture and monitor the EZ best, meaning that often there may be relatively more electrodes. Could this difference in spatial sampling have impacted the results reported for the EZ? Please comment.

- It is stated that the recordings were performed during periods with eyes closed. How was sleep ruled out during these times or, more specifically, how did the authors make sure that the same vigilance state across patients and conditions was maintained? Was sleep scoring performed? Different states of vigilance are known to severely impact synchrony as well as Hurst exponents (e.g. <https://doi.org/10.1073/pnas.1312848110>)

Minor:

- Line 121, 150: LRCTs

- The authors state that "Our results suggest that the co-variability between synchronization levels and LRCT strengths in individuals can be best explained within a revised brain criticality framework." The authors may please want to consider citing other recent and older work that has similarly demonstrating that spatial (i.e. synchronization) and temporal correlations are deeply linked and co-vary based on the criticality framework, e.g. DOI: 10.21203/rs.3.rs-1761325/v1

Reviewer #2 (Remarks to the Author):

The study by Fuscà and colleagues presents a very impressive combination of modeling and experimental testing of a very fundamental hypothesis on human brain function: how do overall activity propagation dynamics analyzed for their criticality relate to the coordination of brain rhythms, which synchronize across variable distance? At first glance this hypothesis looks circular and may represent only two sides of the same coin. However, having deeper insights into brain states and how they self-organize, it is not at all trivial that rhythms at different frequencies coordinate with similar/same rhythms in different brain regions at the same time as rhythms at other frequencies are involved in different processes. And, this is a highly relevant issue, not only in the scenario the authors address in their study by showing that epilepsy is a condition in which exactly this, the criticality regulation goes wrong, much rather the combination of criticality measures and oscillatory brain states will be highly informative for other medical/neurological conditions in which brain function is not healthy. In the case of epilepsy, instead of a well-controlled excitation balanced with just enough inhibition for activity to spread and organize in a constructive fashion, circuits and networks get supercritical (i.e. release more avalanches than helpful) and degrade information processing, not only during seizures, but also interictally. A key development in modern medicine and certainly also in neurology/psychiatry, diagnosing and treating brain diseases, aims at individualized diagnostics and treatments, for which sensitive analyses of complex dynamics are really needed to really make progress. In the case of diagnostic procedures for epilepsy, they have well improved over the last decades, however, there are numerous classes of conditions like sleep and vigilance disorders, intractable pain or tinnitus, to name a few, for which much weaker tools are available, often because pathogenesis is far less clear or even lacking. Therefore, establishing a reliable general method, which can objectively quantify functional pathological deviation of the dynamical behavior of neural processes underlying higher brain function like cognition and mood regulation, is a priority which is very well supported by the results of the study under review, in particular by the results on page 8/9 referring to Figs. 2+3.

Before I can support acceptance of this manuscript, a number of issues have to be taken care of. Also there is a bit too much redundancy with some items, being repeated at least twice. Try to save this space for better explanations of the very complex concepts, results and interpretations! I organized my comments in sections for the sections of the manuscript.

Abstract

The sentence in line 29-31 "Neuronal systems have been postulated to operate near a critical transition point or in an extended regime between disorder (subcritical) and order (supercritical phase) characterized by moderate synchronization and emergent power-law long-range temporal correlations (LRCTs)." contains a fundamental error: disorder, here represented by epilepsy shows supercritical dynamics, while healthy normal brains tend to show subcritical dynamics.

Introduction

The introduction is not well written and needs considerable improvement for clarity and guiding more naïve readers. I will list and describe items which I stumbled over or got stuck, often proposing different wording. However, it will not be enough to simply take over these suggestions as there is considerable work on clarifying and improving readability (true for the entire ms). Second sentence (line 71ff): "... brain function can be achieved ..." sounds like artificial trimming of brain function. Given that the brain is a self-organizing system, a statement like "Healthy brain function expresses in moderate levels of synchronization, while inadequate or excessive synchrony is characteristic of several brain disorders causing functional deficits". Along the same thoughts, "synchronized oscillations" as used in the first sentence of the introduction reflect a very static phenomenon, while "synchronizing oscillations" again reflect more the self-organizing aspect.

Given the wide variety of cognitive performance, it is not surprising that among healthy subjects there is considerable variability in mean levels of synchronization in large scale brain networks. The last sentence of the first paragraph is a very meager statement: I would prefer if the authors could either leave it away (the next sentence anyway provides their preferred factor) or list a few intelligent options, possibly addressing different neurobiological mechanisms by which synchronization can in principle be dynamically modulated!

For a less well knowledgeable reader, the following introduction of brain criticality should be significantly simplified. I find the "explanation" with "order" and "disorder" too simple and not really helpful for brain processes. In addition, "disorder" could imply disease, which is not really the association you want to trigger here!

The next paragraph (line 82ff) starts with "... such a critical point ..." after it has only been mentioned and not well explained. Then there are several different levels and concepts mixed addressing power law, large scale synchroni-zation and functional benefits. Only real insiders of this research area can decode this. In a nature paper you need to deliver text which allows for a simple step-by-step explanation, otherwise the broad readership will become desperate. In the next sentence I do not understand why temporal correlations need "the experimental level" for being indexed, even worse: the "these" cannot unambiguously be related to any of the items in the previous sen-tence: do you refer to the correlations or the mean levels or the functional benefits? Style: ... suggest... indeed ... as If this link would profit from the enumeration. I think that your current results are one of the best pieces of evi-dence that oscillations are tightly linked to critical dynamics.

Line 108ff: "We postulate here that rather than operating near a critical point, healthy brains operate on the subcritical side of an extended critical regime characterized with a diversity of individual operating points." could be improved by adding "... near a single critical point...". The next sentence judges the consequences of a subcritical regime in normal brains. The association between functional benefits of a critical regime and the prevention of excursion to the supercritical/epileptic side suggests causality which is not true. If the authors want to explain the prevention of supercriticality then they should give a convincing argument (or at least formulate a convincing hypothesis)!

The next paragraph begins with "If this were the case..." establishing a false link between your hypothesis for Sync and LRTCs and the wavy judgment and the prevention. Why? Force yourself to spell out logical relations, e.g. like this: "IF healthy brains operate in a mostly subcritical regime, the individual levels of interareal oscillatory synchronization should positively correlate with LRTCs, both across individuals as well as across brain areas." The rest of this paragraph is concrete and clear!

Methods

It would really help the reader to have a short overview of all methods at the beginning of this section rather than at the end of the intro, where hypothesis and main results should be matched! Do not start with the detailed description of the modeling, which plays a minor role... much rather try to organize your methods according to the importance of the methods for achieving the main results.

I am surprised to see that the frequency bands used for analysis e.g. in Fig.3 leave out the band between 30 and 40 Hz which for numerous EEG / MEG studies is the only region actually being used (which is of course even worse than using huge frequency ranges in the gamma-bands above). Of course I checked in Suppl.Fig.1 which clearly shows the existence of signal between 30 and 40 Hz... though ending up in different clusters. I do not mind how the authors address this issue, but it simply needs clarification somewhere. You could add a footnote at "ly 40-65 Hz;" and say why you omit signal below 40 Hz.

Results

start with a description of modeling results is an option, but you could also start with the physiology and provide modeling results when you have described the respective findings. The first result section describes expectations for results in a neural oscillator model, which has reasonable dimen-sions in comparison to the biological system: the Kuramoto model (Simola et al., 2022) works with 100 regions containing 500 oscillators each. What is the rationale for 100 areas? Brodmann described 47, Nieuwenhuys de-scribed 182 (10.1007/s00429-016-1228-7) cortical fields or areas, both parcellations based on state of the art techniques at their time. Later in the results, parcellations of 400 cortical parcels are reported! How does this match the modeling efforts? IF a significantly smaller number of areas with a most likely highly underestimated number

of oscillators (small clusters of pyramids with their local PV+ interneurons) are the basis of the results, the authors should clearly state how the results will be biased by a smaller scale network. The use of real DTI results for estimating connectivity strength between cortical regions helps qualifying this approach, however, it is not clear which DTI data were used: individual / group mean or any human brain connectome data?

Line 148/9 "The brains are thought to operate in the subcritical side of the critical point (Priesemann et al., 2014)." Is a bit bumpy sentence and not sufficiently direct, as the cited paper provides results on many species, signal types and individuals! In the context of this section, a modified statement would be more convincing: "Normal human and non-human brains have been shown to operate more on the subcritical side of the critical point (Priesemann et al., 2014)."

The second result paragraph describes that cortical oscillatory synchronization at the whole brain level and LRCT exhibit large individual variability in MEG/SEEG signal is reported for a really impressive number of sessions/participants, far beyond standards of typical papers! This paragraph contains several sentences (line 161ff/169ff), which read like methods, arguing why this and that. This could be moved to the general overview of the methods section and you here you fully concentrate on the actual results! The result in Fig. 1F for SEEG shows a huge peak in the ripple-gamma range which does not "end"! I would like to see, at what frequency avgDFA comes back below ~ 0.8 !

The third result paragraph reveals one of the most important findings: Synchronization and LRTCs are positively correlated across individuals. The description is clear and relates to the modeling results, and more importantly quantified test/ retest reliability. Griffiths phase needs to be better introduced and explained! A stylistic suggestion for the "on the other hand" in line 201, this should be preceded by a "on the one hand" at the beginning of the same sentence.

The fourth and fifth paragraph describes various linear and quadratic correlations of synchronization and LRTCs in a systematic way also referring to the spatial specificity across brain regions.

Discussion

Line 280 "... et al., 2012), resolving the underpinnings of individual synchronization levels is important." is too much of understatement and far too weak in order to constitute the main motivation for such a complex and demanding study! I request that the authors carefully chose among statements like ", resolving the brain state and thus network conditions of individual synchronization levels is essential for sufficient comprehension that can support novel approaches to future treatments." or "".

Discussing the extended critical regime, the question arises, whether the brain as a whole, which is dominated of activity from the neocortex in MEG and SEEG recordings, is a single system operating between sub- and supercritical modes or whether there are multiple subsystems like the multiple functional networks which each have their own criticality operating point which smears out into a region (Griffiths phase) when observed in combination. The authors only touch on this issue, when in line 303 they speculate that "such Griffiths phase could result from inhomogeneity in regional dynamics, network topology or the combination of both in the brain..." and only refer to Munoz 2010 which is certainly an important theoretical reference, but they should at least refer to their own findings documented in Fig.1

Conclusion

"Our results demonstrate that variability in synchronization levels is regulated by the individual position along an extended critical regime so that healthy brain areas operate in its subcritical and epileptogenic areas in the supercritical side." suggests a causal influence among parameters (here activity spread as revealed by criticality analyses -> oscillatory synchronization), which correlations as reported as the main results of this manuscript can by no means provide. Furthermore, how can a point in a dynamics regime regulate something? A point may determine a process in a different space, but regulation always implies a range of (at least one) parameter(s) which can influence other parameters possibly in a systematic relation and therefore regulate the latter. I am certainly not against an interpretation of their results which tries to provide a concrete and plausible relation among parameters, though direction of influence remains an issue. It may even be an hen-or-egg problem when trying to relate oSYNC and LRCTs, but here the authors could use their frequency-specific information on network dynamics as demonstrated in Fig. 4C

which could provide a basis for suggesting that faster (e. g. gamma-osc based) processes are more likely to reflect feed-forward transmission at least in sensory systems, while slower (in particular alpha- and beta-osc based) processes are an important carrier signal for feedback, thus providing topological information at different times, which can in principle be interpreted as a directed interaction among brain regions.

Figures

I very much appreciate that Fig. 1 contains basic info on topography and signal parameters, even showing a tiny section of raw / broadband filtered signal. The SEEG topo plot in A is weird, suggesting that something went wrong with the transformation of coordinates onto the spherical brain surface.

Minor

Line 273 Martinello et al., 2017; Moretti & Muñoz, 2013, needs a ")" after 2013

Reviewer #3 (Remarks to the Author):

In this manuscript the authors explore the possible connection between the level of synchronization across brain regions in resting-state brain activity, the level of long-time temporal correlations (LTTC) and the individual-dependent distance to criticality in a set of individuals, both healthy (using non-invasive MEG recordings) and patients with drug-resistant epilepsy (using SEEG recordings).

The underlying hypothesis is that the brain can have a "working" point at varying distances from criticality, while keeping functionality (something called Griffith phase), so that by quantifying the distance to the critical point it is possible to objectively discern whether the patient (or a specific brain region within her/him) is healthy, i.e. subcritical, or has entered an epileptic, i.e. supercritical, state. This reminds me of, e.g., a recent paper:

Ezaki, Takahiro, et al.

"Closer to critical resting-state neural dynamics in individuals with higher fluid intelligence."

Communications biology 3.1 (2020): 1-9.

in which the authors find that the distance to the critical resting-state dynamics can gauge the level of fluid intelligence. More in general, the so called criticality idea, is gaining pace in the contexts of both neuroscience and computational neuroscience, so, in principle, this manuscript might be of relevance for the community of researchers working in such fields.

Nevertheless, I find a number of problems and weakness in the present version of the manuscript that make me think that it needs to be improved and that it might not be appropriate to be published in a journal such as Nat. Comm.

In what follows I list some of the existing weakness (both major and minor):

1) Weakness of the computational model:

The computational model is extremely simple, actually it is more a toy model than an actual model attempting to capture the real dynamics. While I appreciate that there are two hierarchical levels in the structural design of the networks, the dynamics is way too oversimplistic. Actually, if I'm not mistaken the model is employed only to show that while the overall level of synchronization grows monotonously with the coupling strength, the LRTC exhibit a peak, which --in analogy with a susceptibility in statistical mechanics-- is assumed to identify the critical point. Question: is it really K what is changing or is the inter-region coupling L ? (please verify). In any case, I would appreciate to see the dependence on both parameters to better grasp the system's phenomenology. Also, I was wondering why there is no structural disorder or across-compartment variability in the model. Given that what the authors aim at studying is precisely the possible variable distance to criticality across brain regions, these should be included in a model, even if it is parsimonious. It would be very illuminating to see the mode, actually having diverse LRTC at different nodes. Related to this, it would also be nice to see the Griffith phase (GP) in the model. If empirical data (allegedly lying in a GP) are to be compared with a model, at least the model should have a GP and it should be quantified, somehow.

In summary: I see the modelling effort as exceedingly naive and not properly corresponding to the empirical measurements/findings, so it turns out to be of very little utility.

2) Dependence of frequency band:

I find it extremely obscure and confusing all the discussion on how results depend on the considered frequency band: some effects such as large synchronization or strong correlations appear in some bands but not in others, and these change from one quantity to another. At the end of the day I remind strongly confused about what is the role of criticality (and the distance to criticality) in each band? For instance, I think the reader can have a hard time trying to make sense of the peaks and valleys in the second row of Fig.2 (or the first and third in Fig.3). Are some there bands closer to critical to others? which ones? Are there bands that are definitely not critical?

The computational model does not help at all to clarify this as there is a well defined frequency interval, or not? can the model be extended to describe broadband signals? In summary, all analysis and discussion on the band-frequency dependence should be profoundly clarified.

3) The measurement of synchronization is rather naive and neglects, e.g., the fact that synchronization can occur in a delayed way (e.g. if there are travelling waves). How do enhanced estimators of synchrony affect the results? Also, the way to infer the distance to criticality --by estimating the degree of quadratic correlation after removing linear ones-- seems not very accurate/convincing to me. In particular the plots in the third row of Fig.3. are a mess from which overall trends are barely discernible. I think the reader would be much more convinced if a single number assigning a distance to criticality to each individual (or brain region?) could be assigned (other studies succeed at doing so).

4) I personally dislike calling "connectivity strength" to the level

of nodal or global synchronization. Actual (physical) connectivity is one thing and the level of correlation/synchronization is another. Of course, this is a semantic issue, but I believe many readers would share my confusion with the used of connectivity to refer to synchronization.

5) In general all plots are hardly illuminating. I do not see "by eye" simple trends that definitely convince me that we are seeing some real trends. This fact, mixed with the confusion between the different bands and the two different types of recordings (that segregate the patients already in two groups and makes the comparison between them doubtful), makes me not feel persuaded at all by the authors conclusions. I think many potential readers will share this sense of dissatisfaction with me.

6) I do not see how Heiney et al. is an adequate reference for the existence of LRTC in vitro.

On the other hand, I would humbly suggest that the authors read and cite the recent review paper

O'Byrne, J., & Jerbi, K. (2022).
How critical is brain criticality?. Trends in Neurosciences

that is very pertinent to the issues addressed here.

In summary, I am not satisfied with the methodologies, analyses performed and clarity of the conclusions so, thus far, I cannot recommend publication of the manuscript in a high-quality journal such as Nat. Comm. At least not in its present form.

Point-by-point replies to reviewers' comments on
"Brain criticality predicts individual synchronization levels in humans"
by Fuscà et al.

We thank the Editor and Reviewers for prompt handling of our manuscript and, in particular, for excellent and thorough comments. Please find below our point-by-point replies (**R**) and actions (**A**). We have done our best to rigorously and comprehensively improve the manuscript along Reviewer suggestions.

Reviewer #1 (Remarks to the Author):

The manuscript provides a report on the relationship between synchronization levels and Hurst exponents in MEG and SEEG. The authors motivate their study by observations from a network model consisting of Kuramoto oscillators which exhibits the well-known synchrony phase transition and a peak in the Hurst exponent at criticality.

The authors argue that these two metrics may consequently be used in empirical data to determine where dynamics resides in phase space, specifically with regards to criticality. The authors report mainly linear positive relationships as well as quadratic relationships between synchrony and Hurst exponent, thus suggesting that networks reside in the subcritical or close to critical regime.

I appreciate the authors' effort to provide a comprehensive study of these two metrics (synchrony, Hurst exponent) across subjects, brain regions and methods (MEG, SEEG).

- **(Reply)** We thank Reviewer #1 for their constructive and insightful comments!

Although one would ideally have the means to tune or perturb brain networks directly and within a subject in order to determine whether there are coherent changes in the two metrics, the present study provides a comprehensive effort to investigate these dynamics across subjects. It assumes that there is a regime where both metrics change in predictable ways.

- **(R)** We fully agree that a perturbational study would be ideal for assessing the co-variability of brain dynamics measures along a manipulation of the control parameter. This is unfortunately beyond the scope of the present paper, but remains a topic we are, in fact, currently addressing both in a follow-up study in humans and in rodents.

There seem to be some frequency bands that behave opposite to what is reported as the main finding. For example, there are negative correlations present in low and high gamma bands in SEEG. How can this be explained and reconciled under the present framework? Is the definition of criticality and where it occurs potentially frequency dependent? Please elaborate more on this as it is central to your interpretation of the findings.

- **(R)** We thank the reviewer for this excellent question. First, we would like to point out that at the whole-brain level, these negative correlations in the SEEG gamma

band were not significant. At the parcel-level, there were indeed also some significant negative correlations in the gamma band but their fraction was so small and close to chance level that we would conclude these negative correlations to not represent a finding robust enough to be considered in the present framework.

- **(R)** Nonetheless, both the present and earlier studies suggest that criticality could indeed be a frequency-dependent phenomenon so that oscillations in different frequency bands could exhibit different operating points. The brain is a heterogeneous system consisting of multiple functionally and spectrally distinct networks. Heterogeneity in structural connectivity has been shown to lead to the emergence of a Griffiths Phase (Muñoz et al., 2010). Our data now extend this by showing that also distinct synchronization networks vary in their operating points, i.e., in their position in the critical regime. This is biologically well plausible as different frequencies have different underlying biological mechanisms and likely also partially distinct control parameters.
- **(Action)** We have now added a sentence to the Results section to describe the results of the negative correlations. We have also added a sentence to the Discussion on the variability of operating points across different frequency bands.

We have added the following changes:

Results, p. 10, lines 21-26:

The spatio-temporally widespread co-variability between synchronization and LRTCs thus further supports the idea that also large areas of the human neocortex are more likely to operate in a Griffiths Phase critical regime instead of at a fixed critical point. The anatomical heterogeneity and frequency-specificity of these anatomical patterns also support the hypothesis that different brain structures or functional systems may express distinct and partially independent operating points (Zhigalov et al., 2017).

Discussion, p. 14, lines 1–18:

We observed variability in the strength of correlations not only across brain regions but also across frequencies, indicating that brain functional networks have a small variability in their operating points in line with the GP framework where the GP arises due to functional and structural heterogeneity (Muñoz et al., 2010). The correlations were linearly positive at the whole brain level and in most parcels and frequencies. However, in alpha, beta, and high-gamma bands there was also a quadratic trend, showing that in a subset of subjects, these functional networks operated around the “peak” of the extended critical regime. Given the fundamental functional roles that alpha and gamma-band oscillations are thought to play in feed-back and feed-forward information processing (Bastos et al., 2015; Jensen et al., 2015; S. Palva & Palva, 2007, 2011; Samaha et al., 2020), respectively, it is conceivable that these oscillations operate around the peak of the putative critical regime, enabling maximal dynamic range and flexibility. The localization of strongest alpha-band correlations into dorsal visual areas and PFC and gamma-band correlations into the ventral stream areas is in line with previous findings of the differential localization of power and synchronization in these frequencies (Honkanen et al., 2015; Klimesch et al., 2007; Popov et al., 2018; Singer, 1999) and with their biologically differential dependence on anatomy, neurobiological mechanisms (Simola et al., 2022; Vezoli et al., 2021; Vinck & Perrenoud, 2019), and neuromodulatory genes. These

findings of heterogeneous neuronal dynamics are also in agreement with previous study from awake mouse visual cortex where scale-free neural activity was limited to specific subsets of neurons (Jones et al., 2023). Heterogeneity across frequencies and functional neuroanatomy could provide a means for adapting behaviour according to environmental demands..

Negative quadratic correlations indicative of close proximity to a critical point seem to be more present in MEG, less so in SEEG (e.g. Fig. 2 E-H, Fig. 3 E-F), if I understand the authors correctly. An important difference between MEG (which here is performed on healthy subjects) and SEEG (which is performed on epilepsy patients) is that patients with SEEG are consequently mostly under antiepileptic drug (AED) treatment. AEDs have been shown to effect LRTCs, long-range spatial correlations and synchrony in predictable ways, i.e. suggestive of a shift towards subcritical dynamics (e.g. <https://doi.org/10.1073/pnas.1911461117>). Have the authors taken AED into account in their analyses? Is it possible that the more linear (and less quadratic) relationship observed in SEEG may similarly indicate a more subcritical state, potentially driven by AEDs?

- **(R)** We thank the reviewer for pointing this out. We fully agree that AEDs constitute one plausible explanation for epileptics operating closer to subcriticality than healthy controls. The patients in this study were all taking AEDs but with considerable variability in the exact compounds and dosage.
- **(A)** We have now clarified the medication status in the Methods section and appended a new Supplementary Table 1 to describe the AED medication. Unfortunately, the technicians performing the SEEG recordings did not keep a record of the gap between the last medication and the resting-state recording time. Due to this and due to heterogeneity of the AED medication, it is not possible to properly investigate the effect of AEDs in our study as the reviewer suggests.
- **(A)** We have also added to the Discussion a sentence about how the lack of quadratic correlations in SEEG could originate from the effects of AEDs (citing the Meisel 2020 paper).

We have added the following changes:

Methods p. 22, lines 5–8:

All patients were taking antiseizure medications (antiepileptic drugs (AEDs) with a large variation in the dosage and compounds (See Supplementary Table 1 for the dosage administered in the morning of the day of the recording), and the time elapsed the last drug administration and the SEEG resting-state recording was not controlled.

Discussion p. 15, lines 8–15:

“ Anti-epileptic drugs, used also by the patients in this study, have been shown to push the brain towards subcriticality (Meisel, 2020; Müller & Meisel, 2023) and may thus constitute a likely explanation for this phenomenon. Another possible contributing factor might be a long-term neuronal plasticity that has globally enhanced inhibitory connections in order to compensate for locally increased excitability in the epileptogenic network, which would drive these healthy brain regions towards the sub-critical side of the critical regime. This is supported by the presence of negative correlations as indicative of supercritical dynamics in gamma-band in some brain areas suggesting an imbalance in the system dynamics.

The authors discuss Griffith spaces and see their results as support. Specifically, they state: “To the best of our knowledge, these findings thus constitute the first empirical evidence towards the discovery of Griffiths-phase-like dynamics in the brains in vivo.” I am not sure how the data provided may prove or disprove these as the present study mainly demonstrates covariation of different types. It has been shown that dynamics, including synchronization levels and related criticality measures, fluctuate widely across patients, regions and time, e.g. <https://doi.org/10.1073/pnas.1513716112>, Fig. 3. Could the authors please elaborate their point more and how it differs from these earlier findings and/or transcends them?

- **(R)** To the best of our knowledge, the present manuscript is the first to uncover the covariance in long-range synchronization and LRCTs. In the present revision, we have elaborated our argument of why such covariance may only be observable in an extended critical regime like the Griffiths phase rather than around a classical critical point.
- **(A)** Nevertheless, we have now written the text more carefully to statements that reach beyond the evidence presented here. We have removed the sentence that the reviewer suggested and fully rewritten the discussion. We hope that the text is now accurate both with respect to prior art and the significance of the present findings.
- **(A)** We apologize for not comprehensively citing the prior art. We have now added a citation to this work that showed co-variation evoked activity and synchronization in two participants with ECoG/EEG sensors and proposed that the co-variation is linked to intrinsic excitability measures tracking antiepileptic drug action. Our work extends this in several fronts: 1) We show co-variation of spontaneous LRCTs of oscillation amplitudes - hallmark of brain criticality - and inter-areal synchronization, extending previous findings of sub-second time-scale correlations of local and large-scale dynamics. 2) We show this co-variation in two extensive datasets, MEG data of healthy human subjects, and SEEG data from patients with epilepsy.
- **(A)** We have made the following edits to explicitly explain the hypothesis regarding the co-variability of LRCTs and synchronization along the GP. We also explicitly write the difference in comparison to previous studies.

Introduction p. 5, lines. 8–22:

However, despite extensive lines of research on inter-areal synchronization and critical brain dynamics, there is only sparse evidence linking the strength of inter-areal correlations with individual critical dynamics in the human brain (Müller & Meisel, 2023) and no experimental papers addressing inter-areal synchronization of oscillations in this context. Moreover, recent theoretical studies suggest that because of diverse structural and mechanistic heterogeneities, neuronal systems are unlikely to operate at a singular critical point (Muñoz et al., 2010). Instead, neuronal systems have been proposed to operate in an extended regime of critical-like dynamics known as the Griffiths phase (GP).

A GP is characterised by power-laws extending over broad regions in parameter space and thus the stretching of a critical point into a wider critical regime (Fig. 1b). In modelling

studies, the intrinsic heterogeneity in the white-matter structural connectivity linking human brain areas lead to the emergence of a GP (Moretti & Muñoz, 2013). While there is only little experimental brain data to support this hypothesis, it is in line with findings showing that different brain systems exhibit partially independent operating points (Zhigalov et al., 2017). The notion of GP in brain dynamics thus implies that instead of there being a single critical point that yields optimal brain functioning, there is a wider range for possible operating points, all of which yield critical-like dynamics and the consequential functional benefits.

We have fully rewritten Discussion p. 12-13.

We are now citing the previous studies in the Discussion, p. 13, lines 5–12:

A subset of the healthy subjects contributed also to a quadratic relationship between synchronization and LRTCs similarly to prior observations of quadratic correlation between oscillation amplitudes and LRTCs in human MEG and EEG (Bruining et al., 2020), in intracranial EEG (Meisel et al., 2015), and in in vitro responses to the manipulation of the E/I balance (Shew et al., 2011; Shew & Plenz, 2013). Our findings also extend previous findings showing covariation of evoked activity and synchronisation in electrocorticography (Meisel et al., 2015), and systematic covariation in decay of spatial and temporal correlations in the human cortex that suggested that this covariation is linked to intrinsic excitability measures tracking antiepileptic drug action (Müller & Meisel, 2023).

The effect of interictal activity: The authors state that channels with interictal discharges were removed from the analysis. How was this done for the channels in the EZ, as the EZ by definition contains most of these discharges? How did this removal impact the results?

- **(R)** we apologize if the description was unclear in the previous version. We did not exclude any contacts due to the presence of spikes – we only excluded “bad” contacts that showed clear non-biological activity during preprocessing. We excluded from the nEZ analyses those subjects (n=8) with more than half of the recording contact marked as epileptic.
- **(R)** Here, we were only interested in studying the synchrony and LRTCs in spontaneous brain activity rather than the pathological interictal events. Therefore, we removed the spike events from assessing synchrony and DFA for EZ and nEZ contacts using identical criteria. The procedure aimed at identifying time-windows with instantaneous large amplitude increases across multiple channels that is known to inflate the PLV (Palva et al., 2018) and DFA (Hardstone et al., 2012) estimates. Thus we haven't removed isolated inter-ictal spikes in single time-windows in single-channels.
- **(R)**The effects of the spike exclusion have been comprehensively tested and documented in our previous work (Arnulfo et al., 2020). Briefly, we partitioned the signals into adjacent time windows of 500 ms and decomposed the signals into 24 frequency bands with Morlet-wavelet filtering. With this time-frequency decomposition, we first computed the mean and the standard deviation of the signal amplitude for each individual SEEG grey matter electrode contact and frequency. A given 500 ms time window was considered putatively contaminated by epileptic (high-amplitude and spike-like) artifacts and rejected if $\geq 10\%$ of contacts exhibited

an amplitude greater than their mean amplitude plus 5 times the standard deviation and this effect was found in $\geq 25\%$ of the frequency bins.

- (R) We previously showed in this cohort that removing time-windows with sharp amplitude increase would not significantly affect PLV estimates (see *Figure for reviewers R1* below which can be found in Suppl. Fig. 3d of Arnulfo et al., 2020 Nature Communications). For the DFA results, the occasional segments with interictal epileptiform spikes or sharp waves were replaced with interpolated data from all channels. After the removal, 96.4 percent of data remained for the analysis and the missing samples were replaced with simulated data generated by a multivariate autoregressive model that retained statistical properties similar to the "clean" time windows.

Figure for reviewers R1: Spike correlation in SEEG data as a function of frequency. Taken from Arnulfo et al, 2020.

- A) In order to clarify how we conducted Ez and nEz analyses, as well as how we handled the spike-like events, we have added the following edits:

Methods, p. 22, lines 18–26:

Subjects who had more than half of their contacts in the EZ (8 subjects) were excluded from the analysis of non-EZ data and included to EZ data. For non-EZ data, we included data from 57 subjects, with a total of 4453 non-EZ contacts (average per subject 78 ± 19 , range 41-124). For EZ, we used only the contacts defined as being epileptogenic – i.e., contacts that were located within the EZ or were part of the seizure propagation network and all contacts from 8 subjects in whom $> 50\%$ contacts had previously been classified as EZ. Subjects who had < 11 EZ contacts were excluded. Thus, both analyses had 57 subjects,

with 49 common to both non-EZ and EZ analyses, 8 included only in non-EZ, and 8 only included in EZ analysis. The total number of EZ contacts was 1725 (average per subject 30 ± 17.5 , range 11-79).

Methods, p. 23, lines 5–11:

Briefly, we partitioned the signals into adjacent time windows of 500 ms and decomposed the signals into 24 frequency bands with Morlet-wavelet filtering. With this time-frequency decomposition, we first computed the mean and the standard deviation of the signal amplitude for each individual SEEG grey matter electrode contact and frequency. A given 500 ms time window was considered putatively contaminated by epileptic (high-amplitude and spike-like) artifacts and rejected if $\geq 10\%$ of contacts exhibited an amplitude greater than their mean amplitude plus 5 times the standard deviation and this effect was found in $\geq 25\%$ of the frequency bins.

Were synchrony levels overall higher (and Hurst exponent levels overall lower) in EZ as would be assumed by this picture or framework that authors propose? If not, how can this be reconciled?

- **(R)** We now show this in a new Figure panel 4C. As the reviewers correctly suggested, synchronization values were higher for the EZ than non-EZ contacts. In contrast, the Hurst DFA exponents were not significantly different between groups of contacts.

Synchrony levels may depend on the spatial sampling, i.e., be higher when electrodes are placed more closely to each other. In SEEG, it is often attempted to capture and monitor the EZ best, meaning that often there may be relatively more electrodes. Could this difference in spatial sampling have impacted the results reported for the EZ?

- **(R)** As suggested by the reviewer, we investigated the possibility that there is a difference in the sampling between EZ and non-EZ contacts. However, there was no statistically significant difference in the sampling between the EZ and non-EZ contacts. This is because the SEEG is not only intended to localizing the EZ area, but also to map the full propagation network of ictal activity. Thus the contacts are equality distributed and channel pair distances within EZ are not different from those outside the EZ (see *Figure for reviewers R2*). We now address this in the Methods and Results sections.
- **(A)** We have added the following edits:

Methods p. 22. Line 26.

The average distance between EZ and non-EZ contacts was similar (Supplementary Figure 8).

Figure for reviewers R2: Distribution of distances between contacts, for pairs either both within or both outside epileptic zone (EZ). Equivalent to Suppl. Fig. 8.

It is stated that the recordings were performed during periods with eyes closed. How was sleep ruled out during these times or, more specifically, how did the authors make sure that the same vigilance state across patients and conditions was maintained? Was sleep scoring performed? Different states of vigilance are known to severely impact synchroSny as well as Hurst exponents (e.g. <https://doi.org/10.1073/pnas.1312848110>)

- **(R)** This is an important consideration and a challenge especially in clinical settings, which we aimed to account for with a dedicated experimental protocol. The acquisition montage used at Niguarda hospital typically includes two scalp derivations, which enables sleep staging. We used EEG channel-pair (C3-P3) to quantify delta amplitude profiles. Thus the data analyzed in this work were selected by trained technicians from a 24h long period, excluding periods with electrophysiological signs of drowsiness or sleep onset.
- **(A)** We have added the following text.

Methods, p. 21, lines 20–21:

Visual inspection of delta amplitude profiles from EEG channel pair C3-P3 and video recordings of the patient were used by trained technicians to ascertain the absence of signs of sleep or drowsiness.

Minor:

- Line 121, 150: LRCTs

- The authors state that “Our results suggest that the co-variability between synchronization levels and LRCT strengths in individuals can be best explained within a revised brain criticality framework.” The authors may please want to consider citing other recent and older work that has similarly demonstrating that spatial (i.e. synchronization) and temporal correlations are deeply linked and co-vary based on the criticality framework, e.g. DOI: 10.21203/rs.3.rs-1761325/v1

- **(R)** We thank the reviewer for the suggestion
- **(A)** We now cite this paper in the Discussion.

Discussion p. 13. Lines 9–12:

Our findings also extend previous findings showing covariation of evoked activity and synchronisation in electrocorticography (Meisel et al., 2015), and systematic covariation in decay of spatial and temporal correlations in the human cortex that suggested that this covariation is linked to intrinsic excitability measures tracking antiepileptic drug action (Müller & Meisel, 2023).

Reviewer #2 (Remarks to the Author):

The study by Fusca and colleagues presents a very impressive combination of modeling and experimental testing of a very fundamental hypothesis on human brain function: how do overall activity propagation dynamics analyzed for their criticality relate to the coordination of brain rhythms, which synchronize across variable distance?

At first glance this hypothesis looks circular and may represent only two sides of the same coin. However, having deeper insights into brain states and how they self-organize, it is not at all trivial that rhythms at different frequencies coordinate with similar/same rhythms in different brain regions at the same time as rhythms at other frequencies are involved in different processes.

And, this is a highly relevant issue, not only in the scenario the authors address in their study by showing that epilepsy is a condition in which exactly this, the criticality regulation goes wrong, much rather the combination of criticality measures and oscillatory brain states will be highly informative for other medical/neurological conditions in which brain function is not healthy.

In the case of epilepsy, instead of a well-controlled excitation balanced with just enough inhibition for activity to spread and organize in a constructive fashion, circuits and networks get supercritical (i.e. release more avalanches than helpful) and degrade information processing, not only during seizures, but also interictally.

A key development in modern medicine and certainly also in neurology/psychiatry, diagnosing and treating brain diseases, aims at individualized diagnostics and treatments, for which sensitive analyses of complex dynamics are deadly needed to really make progress.

In the case of diagnostic procedures for epilepsy, they have well improved over the last decades, however, there are numerous classes of conditions like sleep and vigilance disorders, intractable pain or tinnitus, to name a few, for which much weaker tools are available, often because pathogenesis is far less clear or even lacking.

Therefore, establishing a reliable general method, which can objectively quantify functional pathological deviation of the dynamical behavior of neural processes underlying higher brain function like cognition and mood regulation, is a priority which is very well supported by the results of the study under review, in particular by the results on page 8/9 referring to Figs. 2+3.

Before I can support acceptance of this manuscript, a number of issues have to be taken care of. Also there is a bit too much redundancy with some items, being repeated at least twice. Try to save this space for better explanations of the very complex concepts, results and interpretations!

I organized my comments in sections for the sections of the manuscript.

Abstract

The sentence in line 29-31 “Neuronal systems have been postulated to operate near a critical transition point or in an extended regime between disorder (subcritical) and order (supercritical phase) characterized by moderate synchronization and emergent power-law long-range temporal correlations (LRTCs).” contains a fundamental error: disorder, here represented by epilepsy shows supercritical dynamics, while healthy normal brains tend to show subcritical dynamics.

- **(Reply)** We thank reviewer #2 for pointing this out and apologize for the mistake.
- **(Action)** We have now fully revised the abstract.

Introduction

The introduction is not well written and needs considerable improvement for clarity and guiding more naïve read-ers. I will list and describe items which I stumbled over or got stuck, often proposing different wording. However, it will not be enough to simply take over these suggestions as there is considerable work on clarifying and improving readability (true for the entire ms).

Second sentence (line71ff): “ ... brain function can be achieved ...” sounds like artificial trimming of brain function. Given that the brain is a self-organizing system, a statement like “Healthy brain function expresses in moderate levels of synchronization, while inadequate or excessive synchrony is characteristic of several brain disorders causing functional deficits”.

Along the same thoughts, “synchronized oscillations” as used in the first sentence of the introduction reflect a very static phenomenon, while “synchronizing oscillations” again reflect more the self-organizing aspect.

- **(R)** We thank the reviewer for their very good suggestions.
- **(A)** We have fully rewritten the introduction and discussion to clarify the theoretical framework and to improve the readability of the manuscript. We have done our best that it is accessible to lay reader. We have now modified also the text referred by the reviewer.

Given the wide variety of cognitive performance, it is not surprising that among healthy subjects there is considerable variability in mean levels of synchronization in large scale brain networks. The last sentence of the first paragraph is a very meager statement: I would prefer if the authors could either leave it away (the next sentence anyway provides their preferred factor) or list a few intelligent options, possibly addressing different neurobiological mechanisms by which synchronization can in principle be dynamically modulated!

- **(R)** We thank the reviewer for this suggestion.
- **(A)** We have now removed the sentence as suggested.

For a less well knowledgeable reader, the following introduction of brain criticality should be significantly simplified. I find the “explanation” with “order” and “disorder” too simple and not

really helpful for brain processes. In addition, “disorder” could imply disease, which is not really the association you want to trigger here!

The next paragraph (line 82ff) starts with “... such a critical point ...” after it has only been mentioned and not well explained. Then there are several different levels and concepts mixed addressing power law, large scale synchronization and functional benefits. Only real insiders of this research area can decode this.

In a nature paper you need to deliver text which allows for a simple step-by-step explanation, otherwise the broad readership will become desperate. In the next sentence I do not understand why temporal correlations need “the experimental level” for being indexed, even worse: the “these” cannot unambiguously be related to any of the items in the previous sentence: do you refer to the correlations or the mean levels or the functional benefits? Style: ... suggest... indeed ... as If this link would profit from the enumeration. I think that your current results are one of the best pieces of evidence that oscillations are tightly linked to critical dynamics.

- (R) We agree that the introduction was not easy to comprehend.
- (A) We have revised Figure 1 (see excerpt below as *Figure for reviewers R3*) to decipher the theoretical framework and the hypothesis in a more accurate way and helps also the lay reader to understand the basic idea of the manuscript.

Figure 1 - NEW

Figure for reviewers R3: Theoretical framework and computational modeling. Excerpt from manuscript figure 1.

a. Concept of critical-like brain dynamics. The state-space of critical dynamics emerges from a position in parameter space and leads to observables such as synchronization and long-range temporal correlations (LRTCs). **b.** Computational modelling with connected Kuramoto oscillators of critical dynamics, using either uniform or realistic heterogeneous connection to simulate classical criticality or Griffiths phase (GP), respectively. PLV and DFA values are shown as a function of the normalized control parameter (nK), i.e., the coupling strength between oscillators. **c.** The differential relationships of PLV and DFA as function of the control parameter for realistic connection weights. More detailed description in manuscript figure legend.

- (A) We have now also fully rewritten the introduction and we hope that it is now clear also for the lay reader.

Line 108ff: “We postulate here that rather than operating near a critical point, healthy brains operate on the subcritical side of an extended critical regime characterized with a diversity of individual operating points.” could be improved by adding “... near a single critical point...”. The next sentence judges the consequences of a subcritical regime in normal brains.

- **(A)** We have now clarified the hypothesis regarding the variability of operating points across individuals and across neuronal subsystems and formulated as suggested by the reviewer.

Introduction p. 5, line 23 – 27:

Here, we build on the framework that healthy brains operate in an extended critical regime, the GP, and posit that the operating points would vary near a single critical point both across individuals and across neuronal sub-systems within individuals. We thus hypothesized that individual variability in the operating points within the critical regime would predict the individual co-variability in the synchronization levels and LRTCs (Figure 1c).

The association between functional benefits of a critical regime and the prevention of excursion to the supercritical/epileptic side suggests causality which is not true. If the authors want to explain the prevention of supercriticality then they should give a convincing argument (or at least formulate a convincing hypothesis)!

- **(R)** We apologise again for bad wording.
 - **(A)** We have now fully rewritten this paragraph and we hope that it now reads well. The new text can be found in Discussion p. 13, line 4 – p. 14, line 2.

The next paragraph begins with “If this were the case...” establishing a false link between your hypothesis for Sync and LRTCs and the wavy judgment and the prevention. Why? Force yourself to spell out logical relations, e.g. like this: “IF healthy brains operate in a mostly subcritical regime, the individual levels of interareal oscillatory synchronization should positively correlate with LRTCs, both across individuals as well as across brain areas.” The rest of this paragraph is concrete and clear!

- **(A)** We thank the reviewer for the suggestion and have corrected this.

Introduction p. 5, line 27 – p.6, line 1:

If healthy brains operate in a mostly subcritical regime as suggested previously (Toker et al., 2022), the individual levels of interareal oscillatory synchronization should be positively correlated with LRTCs, both across individuals as well as across brain areas.

Methods

It would really help the reader to have a short overview of all methods at the beginning of this section rather than at the end of the intro, where hypothesis and main results should be matched! Do not start with the detailed description of the modeling, which plays a minor role... much rather try to organize your methods according to the importance of the methods for achieving the main results.

- **(A)** We have now removed the methods section from the introduction and added a short overview of the methods to the beginning of the Methods section.

Methods p. 18, lines 4-15:

We first used a computational model based on hierarchical Kuramoto-based computational model to generate brain dynamics in GP and in the classical criticality framework and to investigate the relationship between oscillatory inter-areal synchronisation and LRTCs across wide range operating points near the critical point. We then estimated inter-areal synchronization and LRTCs in source-reconstructed resting-state magnetoencephalography (MEG) data recorded from healthy subjects and resting-state stereo-electroencephalographic (SEEG) data recorded from epilepsy patients. Simulated, MEG, and SEEG data were filtered into narrow-band frequency time series. Synchronization was computed between all pairs of brain regions/electrodes and LRTCs were analysed with Detrended Fluctuation Analysis (DFA) for each parcel. To test whether the heterogeneity of the individual variability of LRCTs and synchronisation levels reflect the variability of individual operating points as predicted by the GP framework, we computed linear and quadratic correlations between synchronization and LRTC metrics. In SEEG data, we carried these analyses out separately for EZ and non-EZ networks and computed the difference of correlation coefficients.

I am surprised to see that the frequency bands used for analysis e.g. in Fig.3 leave out the band between 30 and 40 Hz which for numerous EEG / MEG studies is the only region actually being used (which is of course even worse than using huge frequency ranges in the gamma-bands above). Of course I checked in Suppl.Fig.1 which clearly shows the existence of signal between 30 and 40 Hz... though ending up in different clusters. I do not mind how the authors address this issue, but it simply needs clarification somewhere. You could add a footnote at “ly 40-65 Hz;” and say why you omit signal below 40 Hz.

- **(R)** We apologize for this mistake. Indeed, we had not chosen an optimal distribution of frequencies.
- **(A)** We have now re-calculated all results using a better frequency resolution, including 32 and 38 Hz low-gamma. There were essentially no qualitative or quantitative changes in the results with the finer resolution, but this is still an improvement in the quality of the study.

Results

Start with a description of modeling results is an option, but you could also start with the physiology and provide modeling results when you have described the respective findings.

- **(R)** We thank the reviewer for the suggestion. We have considered this, but as we use the model data also to explain the hypothesis in both the Introduction and Results, we do need to maintain the order in which the modelling results are presented in the beginning.

The first result section describes expectations for results in a neural oscillator model, which has reasonable dimensions in comparison to the biological system: the Kuramoto model (Simola et al., 2022) works with 100 regions containing 500 oscillators each. What is the rationale for 100 areas? Brodmann described 47, Nieuwenhuys described 182 (10.1007/s00429-016-1228-7) cortical fields or areas, both parcellations based on state of the art techniques at their time. Later in the results, parcellations of 400 cortical parcels are reported! How does this match the modeling efforts? IF a significantly smaller number of areas with a most likely highly underestimated number of oscillators (small clusters of pyramids with their local PV+ interneurons) are the basis of the results, the authors should clearly state how the results will be biased by a smaller scale network.

- **(R)** The model behavior is universal across large numbers of oscillators and nodes. Hence the model is relatively agnostic to the number of brain regions (*i.e.* to the resolution of the cortical atlas) and the nodes of the model correspond to brain regions *per se* only in the sense of how they are linked by the structural connectome.
- **(A)** For internal consistency of the manuscript, we have now re-computed all computational model simulations with an atlas of 400 regions to match the experimental MEG and SEEG data. We have now modified the figures, results, and methods accordingly.

The use of real DTI results for estimating connectivity strength between cortical regions helps qualifying this approach, however, it is not clear which DTI data were used: individual / group mean or any human brain connectome data?

- **(R)** We apologise for the missing information. We used structural connectomes of the group mean of 57 subjects randomly selected from the WU-Minn 1200 subjects dataset of the Human Connectome Project (HCP).
- **(A)** We have now clarified this in the Methods and added a section to describe Diffusion tensor imaging (DTI) data in p. 19-20, lines 21-7.

Line 148/9 “The brains are thought to operate in the subcritical side of the critical point (Priesemann et al., 2014).” Is a bit bumpy sentence and not sufficiently direct, as the cited paper provides results on many species, signal types and individuals! In the context of this section, a modified statement would be more convincing: “Normal human and non-human brains have been shown to operate more on the subcritical side of the critical point (Priesemann et al., 2014).”

- **(A)** We fully agree with the reviewer and have changed the text accordingly.

Discussion p. 13, lines 1–5:

The present results converge to show that healthy brain areas operate mostly in the subcritical side of the GP, which is in line with previous studies emphasizing the possibility and importance of healthy mammalian brains to operate in a subcritical-like regime. Prior modeling and in vivo studies from normal human and non-human brains have suggested this operating point to be in the subcritical phase very near the critical point (Toker et al., 2022; Wilting & Priesemann, 2019; Xu et al., 2022).

The second result paragraph describes that cortical oscillatory synchronization at the whole brain level and LRCT exhibit large individual variability in MEG/SEEG signal is reported for a really impressive number of sessions/participants, far beyond standards of typical papers! This paragraph contains several sentences (line 161ff/169ff), which read like methods, arguing why this and that. This could be moved to the general overview of the methods section and you here you fully concentrate on the actual results!

- **(R)** We thank the reviewer for the compliment.
- **(A)** We have now removed the method-like text and rewritten this section to highlight the results. Results, p. 7–9.

The result in Fig. 1F for SEEG shows a huge peak in the ripple-gamma range which does not “end”! I would like to see at what frequency avgDFA comes back below ~0.8!

- **(R)** Unfortunately, we can not explore this in our dataset because we are limited by the sampling rate of 1000 Hz of the clinical SEEG data acquisition system. The Nyquist theorem denotes the half of this frequency as the maximum that can be obtained information about, and in practice, filtering requires at least ~3 samples per cycle. This limits us to frequencies below 300 Hz. We do, however, share the Reviewer’s curiosity towards this observation and we are exploring the possibility of acquiring SEEG data with greater sampling rate.

The third result paragraph reveals one of the most important findings: Synchronization and LRCTs are positively correlated across individuals. The description is clear and relates to the modeling results, and more importantly quantified test/ retest reliability. Griffiths phase needs to be better introduced and explained!

- **(R)** We fully agree that the explanation of the Griffiths phase (GP) was not optimal in the previous submission.
- **(A)** We have now implemented several improvements to the explanation of the GP concept to a lay reader.
 - 1) We use the computation modeling to demonstrate the presence of GP and use schematic illustrations of the hypothesis. New figures 1A-C.
 - 2) We have fully rewritten Introduction to better explain the concept of the extended critical regime, the GP, and what are the predictions of the GP. We clarify that the GP predicts that there is a large variability in individual operating points and that this should lead to correlations between LRCTs and synchronization across individuals.

Introduction p. 5, line 11–27:

Moreover, recent theoretical studies suggest that because of diverse structural and mechanistic heterogeneities, neuronal systems are unlikely to operate at a singular critical point (Muñoz et al., 2010). Instead, neuronal systems have been proposed to operate in an extended regime of critical-like dynamics known as the Griffiths phase (GP) (Moretti & Muñoz, 2013; Ódor & de Simoni, 2021).

A GP is characterised by power-laws extending over broad regions in parameter space and thus the stretching of a critical point into a wider critical regime (Fig. 1b). In modelling studies, the intrinsic heterogeneity in the white-matter structural connectivity linking human brain areas lead to the emergence of a GP (Moretti & Muñoz, 2013). While there is only little experimental brain data to support this hypothesis, it is in line with findings showing that different brain systems exhibit partially independent operating points (Zhigalov et al., 2017). The notion of GP in brain dynamics thus implies that instead of there being a single critical point that yields optimal brain functioning, there is a wider range for possible operating points, all of which yield critical-like dynamics and the consequential functional benefits.

Here, we build on the framework that healthy brains operate in an extended critical regime, the GP, and posit that the operating points would vary near a single critical point both across individuals and across neuronal sub-systems within individuals. We thus hypothesized that individual variability in the operating points within the critical regime would predict the individual co-variability in the synchronization levels and LRTCs (Figure 1c). We have added explanation on how the results of variability of individual GS and LRTCs values and specifically of their correlations support the idea of variability of individual operating points and thus the presence of GP.

Results p.9, lines 25–28:

The robust co-variability between the strengths of neuronal synchronization and LRTCs implies that there is a large range of possible individual operating points. This supports our hypothesis that the human brains operate in an extended critical regime, such as the Griffiths Phase, instead of being confined to the vicinity of a singular critical point.

Finally, in the Discussion, we now discuss the results in the context of GP vs. classical criticality models.

Discussion p. 12, lines 10 – 24:

We first showed, using computational modeling of macroscopic oscillation synchronization dynamics, how the operating point along GP determines the distinct modes of co-variation of long-range phase synchronization and LRTCs in local oscillation amplitude fluctuations. We then leveraged the large inter-individual variability across subjects in the LRCTs and synchronization levels to investigate whether they co-vary within individuals and regions in MEG and SEEG data and thereby deduced their in vivo operating regimes. We found that in healthy subjects and in healthy brain areas of epileptic patients, inter-areal synchronisation of neuronal oscillations was positively correlated with LRTCs in all frequencies above 5 Hz in MEG and in the alpha and gamma bands in SEEG (see Fig. 3). In addition, we also observed a significant quadratic trend, which indicates that a subset of subjects operates on both sides of the peak of the critical regime, exhibiting the strongest LRTCs and thereby operating closest to the theoretically attainable criticality (see Fig. 1c).

These findings strongly suggest that individual and regional variability in the synchronization levels is explained by the variability in the operating points in line with the GP framework. The GP framework thus extends the classical notion of criticality being constrained to a singular point in the space of control parameters which underlies models in which the brain is assumed to operate either at the critical point (Beggs & Plenz, 2003), near the critical point (Chialvo, 2010), or in a slightly subcritical regime (Priesemann et al., 2014; Toker et al., 2022; Wilting & Priesemann, 2019).

A stylistic suggestion for the “on the other hand” in line 201, this should be preceded by a “on the one hand” at the beginning of the same sentence.

- (A) We agree with the reviewer here. The whole section has now been rewritten.

The fourth and fifth paragraph describes various linear and quadratic correlations of synchronization and LRTCs in a systematic way also referring to the spatial specificity across brain regions.

- (A) We have now fully rewritten the section in Results for the paragraph p. 10 *Synchronization and LRTCs are positively correlated across brain regions* to be specific about the specificity across brain regions.

Discussion

Line 280 “... et al., 2012), resolving the underpinnings of individual synchronization levels is important.” is too much of understatement and far too weak in order to constitute the main motivation for such a complex and demanding study! I request that the authors carefully chose among statements like “, resolving the brain state and thus network conditions of individual

synchronization levels is essential for sufficient comprehension that can support novel approaches to future treatments.” or “”.

- (A) We thank the Reviewer for this great comment. This section has now been fully rewritten to highlight the relevance of the finding and motivation based on the GP.

Discussion p. 11. Lines 22–25:

Discovering the brain-dynamics basis defining the boundaries of such variability is essential both for understanding the neurobiological factors underlying individual differences in cognitive performance and for developing new therapeutic approaches for brain diseases characterized by abnormal synchronization levels.

Discussing the extended critical regime, the question arises, whether the brain as a whole, which is dominated of activity from the neocortex in MEG and SEEG recordings, is a single system operating between sub- and supercritical modes or whether there are multiple subsystems like the multiple functional networks which each have their own criticality operating point which smears out into a region (Griffiths phase) when observed in combination. The authors only touch on this issue, when in line 303 they speculate that “such Griffiths phase could result from inhomogeneity in regional dynamics, network topology or the combination of both in the brain...” and only refer to Munoz 2010 which is certainly an important theoretical reference, but they should at least refer to their own findings documented in Fig.1

- **(R)** We thank the reviewer for these thoughts. This is a fundamental and very interesting question that, to the best of our knowledge, has not been answered in any prior art. The GP is commonly considered to be a property of the system, and could indeed be seen to arise from the smearing of diverse operating points of the elements (where this diversity is driven by structural heterogeneity that interacts with the control parameter, even though the control parameter value *per se* is the same for each element). A deeper scrutiny of this topic, and in particular the question of how the GP of the system influences the dynamics of the nodes, remains a topic for upcoming research.
- **(A)** We have now extended the discussion about variability of individual operating points in functionally and anatomically separate networks. This variability is in line with the idea of the GP resulting from the inhomogeneity of regional dynamics and anatomy. We are now discussing this explicitly in the Discussion.

Discussion, p. 12, Lines 2–8:

Moreover, we hypothesized that the structural and functional heterogeneities in human brains would lead to the stretching of a theoretical critical point into a regime of critical-like dynamics, the Griffiths phase (GP). This has previously been suggested by modeling studies (Moretti & Muñoz, 2013; Muñoz et al., 2010; Ódor & de Simoni, 2021) and would enable the expression of individual operating points in a wider regime. If variability of in synchronization levels would be explained by the variability in the individual operating points, inter-areal oscillatory synchronization levels should be correlated with LRCTs across both individuals and brain regions.

p.12, line 21 – p.13, line 2:

The GP framework thus extends the classical notion of criticality being constrained to a singular point in the space of control parameters which underlies models in which the brain is assumed to operate either at the critical point (Beggs & Plenz, 2003), near the critical point (Chialvo, 2010), or in a slightly subcritical regime (Priesemann et al., 2014; Toker et al., 2022; Wilting & Priesemann, 2019). Importantly, both here and in our earlier study (Zhigalov et al., 2017), we found heterogeneity in dynamics as well as co-variance of LRCTs and synchronization levels also across neocortical regions within specific frequency bands. This suggests that the different functional brain systems in individual subjects operate in different positions of the critical regime (Zhigalov et al., 2017) rather than being fully controlled by a global parameter.

p.13, lines 15–23:

However, here we posit that a singular operating point in the subcritical phase would be biologically suboptimal because it would be associated with attenuated propagation of neuronal processing, inadequate synchronization, and loss of power-law scaling that endows the functional benefits of criticality. Operation in the subcritical-to-critical side of a GP, on the other hand, yields a range of operating points that are sufficiently far from the supercritical regime that may predispose to epileptogenesis (Meisel et al., 2015; Monto et al., 2007) while benefitting from the functional advantages that have been ascribed to operation at the classical critical point, such as maximal dynamic range, maximal capacity for information transmission and storage, and optimal computational efficiency (Beggs, 2008; Chialvo, 2010; S. Palva & Palva, 2018; Shew & Plenz, 2013).

Conclusion

“Our results demonstrate that variability in synchronization levels is regulated by the individual position along an extended critical regime so that healthy brain areas operate in its subcritical and epileptogenic areas in the supercritical side.” suggests a causal influence among parameters (here activity spread as revealed by criticality analyses -> oscillatory synchronization), which correlations as reported as the main results of this manuscript can by no means provide.

Furthermore, how can a point in a dynamics regime regulate something? A point may determine a process in a different space, but regulation always implies a range of (at least one) parameter(s) which can influence other parameters possibly in a systematic relation and therefore regulate the latter. I am certainly not against an interpretation of their results which tries to provide a concrete and plausible relation among parameters, though direction of influence remains an issue.

- **(A)** We fully agree and have now fully rewritten the conclusion section.

p. 16, lines 1–7:

Synchronisation of neuronal oscillations is fundamental for brain function, but synchronization levels vary largely across individuals. The present results demonstrate that synchronization levels and LRTCs are correlated across subjects in a manner which suggests that human brains operate in the subcritical side and around the peak of an extended critical regime, the Griffiths Phase (GP), while epileptogenic areas operate in the supercritical side. This implies that the variability in synchronization levels is determined by the individual operating point, i.e., by the individual position in the GP.

It may even be an hen-or-egg problem when trying to relate oSYNC and LRCTs, but here the authors could use their frequency-specific information on network dynamics as demonstrated in Fig. 4C which could provide a basis for suggesting that faster (e. g. gamma-osc based) processes are more likely to reflect feed-forward transmission at least in sensory systems, while slower (in particular alpha- and beta-osc based) processes are an important carrier signal for feedback, thus providing topological information at different times, which can in principle be interpreted as a directed interaction among brain regions.

- **(A)** We thank the reviewer for this interesting suggestion. We have now added a discussion about oscillations in different frequency bands in coordinating feed-forward and feed-back processing. However, as this study is based on resting-state data, which can't be used to directly support these hypotheses, we have left this discussion rather speculative.

Discussion p. 14, lines 1–10:

We observed variability in the strength of correlations not only across brain regions but also across frequencies, indicating that brain functional networks have a small variability in their operating points in line with the GP framework where the GP arises due to functional and structural heterogeneity (Muñoz et al., 2010). The correlations were linearly positive at the whole brain level and in most parcels and frequencies. However, in alpha, beta, and high-gamma bands there was also a quadratic trend, showing that in a subset of

subjects, these functional networks operated around the “peak” of the extended critical regime. Given the fundamental functional roles that alpha and gamma-band oscillations are thought to play in feed-back and feed-forward information processing (Bastos et al., 2015; Jensen et al., 2015; S. Palva & Palva, 2007, 2011; Samaha et al., 2020), respectively, it is conceivable that these oscillations operate around the peak of the putative critical regime, enabling maximal dynamic range and flexibility.

Figures

I very much appreciate that Fig. 1 contains basic info on topography and signal parameters, even showing a tiny section of raw / broadband filtered signal. The SEEG topo plot in A is weird, suggesting that something went wrong with the transformation of coordinates onto the spherical brain surface.

- We regret that the figure looks weird, but we are not sure which topoplot or issue the reviewer is referring to. There is no transformation and no common FSL-style sphere coregistration, the positions of the electrodes are assigned to parcellations in their individual space. The top image is meant to be just a rendition to show intracranial contacts, though it does show the actual electrode arrays for a single example patient. We checked the location in the individual MRI and they actually could be shown slightly more ventral, so we updated the figure accordingly for the sake of realism, even if it is only for illustrative purposes. See the old topoplot on left and the new topoplot on right in *Figure for reviewers R4*.

Figure for reviewers R4: Old and new plots of SEEG contacts, as in manuscript Figure 1.

Minor

Line 273 Martinello et al., 2017; Moretti & Muñoz, 2013, needs a “)” after 2013

- (A) We thank the reviewer for pointing this out. We have now modified this.

Reviewer #3 (Remarks to the Author):

In this manuscript the authors explore the possible connection between the level of synchronization across brain regions in resting-state brain activity, the level of long-time temporal correlations (LTTC) and the individual-dependent distance to criticality in a set of individuals, both healthy (using non-invasive MEG recordings) and patients with drug-resistant epilepsy (using SEEG recordings). The underlying hypothesis is that the brain can have a "working" point at varying distances from criticality, while keeping functionality (something called Griffith phase), so that by quantifying the distance to the critical point it is possible to objectively discern whether the patient (or a specific brain region within her/him) is healthy, i.e. subcritical, or has entered an epileptic, i.e. supercritical, state. This reminds me of, e.g., a recent paper: Ezaki, Takahiro, et al. "*Closer to critical resting-state neural dynamics in individuals with higher fluid intelligence.*" *Communications biology* 3.1 (2020): 1-9. in which the authors find that the distance to the critical resting-state dynamics can gauge the level of fluid intelligence. More in general, the so-called criticality idea, is gaining pace in the contexts of both neuroscience and computational neuroscience, so, in principle, this manuscript might be of relevance for the community of researchers working in such fields.

Nevertheless, I find a number of problems and weakness in the present version of the manuscript that make me think that it needs to be improved and that it might not be appropriate to be published in a journal such as *Nat. Comm.* In what follows I list some of the existing weakness (both major and minor):

- **(Reply)** We are grateful to Reviewer #3 for their valuable comments and critique of our manuscript. We agree with Reviewer #3 on many occasions and feel that their critique has been instrumental in improving the reach and accessibility of the manuscript.
- **(Action)** We have improved the manuscript in many respects, with all modeling work, experimental analyses, and figure largely redone, as well as with the text rewritten almost completely. We outline these changes below in the point-by-point replies and hope that they resolve the concerns of Reviewer #3 about the problems and weaknesses of this manuscript.

1) Weakness of the computational model:

The computational model is extremely simple, actually it is more a toy model than an actual model attempting to capture the real dynamics. While I appreciate that there are two hierarchical levels in the structural design of the networks, the dynamics is way too oversimplistic. Actually, if I'm not mistaken the model is employed only to show that while the overall level of synchronization grows monotonously with the coupling strength, the LRTCs exhibit a peak, which — in analogy with a susceptibility in statistical mechanics — is assumed to identify the critical point.

- **(R)** We appreciate this comment and fully agree with Reviewer #3 in that the model is indeed 'statistical mechanics'-like in the sense of reproducing the canonical and universal behavior of critical-like systems, where order (here synchrony) increases sigmoidally as a function of the control parameter while variance/entropy (here LRTCs) peaks at the critical point.

- **(R)** We do, however, politely disagree with the notion that the model would not capture real dynamics. While the model is indeed an abstraction of oscillating neuronal assemblies and is not intended to be based on neurophysiologically accurate mechanisms, the model does reproduce quantitatively very well the real, experimentally observable dynamics. Moreover, neuronal synchronization dynamics (rather than, for example, specific synaptic neurophysiological mechanisms *per se*) are the key construct in this investigation and the present model is specifically a model of emergent synchronization dynamics.
- The *Figure for reviewers R5* below illustrates the comparison between measures of model and real dynamics. It is from another manuscript in preparation and we present it here as information for reviewers to address the present comment.
- The dynamics that the model yields are ‘realistic’ both in the sense of exhibiting *in vivo*-like LRTC scaling exponents, mean synchronization levels (*Fig. R5a and b*). Moreover, with model fitting applied with individual structural and functional data, the model yields excellent fits (Pearson correlation up to 0.6) with the specific cortical architecture and network structure of LRTCs and synchronization, respectively (*Fig. R5c*).

Figure for reviewers R5: Comparison of synchronization dynamics values in the model and real data. **a.** Correlation (Pearson correlation coefficient) of the cortical neuroanatomy of inter-areal synchronization (upper panel) and LRTCs (lower panel) between the Hierarchical Kuramoto model and MEG resting-state recording of 11 Hz neuronal oscillations as a function of the control parameters of the model (intra- and inter-nodal coupling K and L). Both Synchronization and LRTCs are correlated between the model and real data but exclusively in the critical region of model dynamics. Please see Figure for reviewers R6 for basic model phenomenology where synchronization increases as a function of K and L while LRTCs peak in the critical range at moderate synchronization. **b.** Synchronization (PLV for model, wPLI for MEG) and LRTCs (DFA scaling exponent) measurements are similarly distributed in modeled and real data. **c.** As a further qualitative demonstration of the realism readily achievable with the model, we illustrate here an individual subject with the model synchronization dynamics (both synchronization and LRTCs) co-fitted to the observed data.

Question: is it really K what is changing or is the inter-region coupling L ? (please verify). In any case, I would appreciate to see the dependence on both parameters to better grasp the system's phenomenology.

- **(R)** The model may be tuned by both the nodal (local) control parameter K and the inter-areal control parameter L . In the current illustrations we keep L constant and adjust K , but the behavior is universal and all model plots would appear qualitatively the same if K was constant and L was illustrated as the control parameter.
- **(R)** We illustrate this model behavior below in *Figure for reviewers R6*, which shows order (synchrony) and LRTCs (DFA scaling exponent) as a function of K and L . The model plots in manuscript Fig. 1 and Fig. 4 are essentially rows cut from such K - L parameter planes.

Figure for reviewers R6: Model dynamics as a function of local (K) and inter-areal (L) control parameters.

Mean node order was obtained as the mean of classical Kuramoto order across nodes. Inter-areal synchronization was quantified with PLV and averaged across all node pairs. LRTCs were assessed with DFA. The DFA exponents peak between the sub- and supercritical phases at moderate order and synchronization, and indicate a regime of emergent power-law scale-free dynamics.

- We have now modified the Method section to describe the model more accurately.

Methods p. 19, lines 8–13:

The phases of all oscillators in a node were averaged to derive a single time series per region, whose absolute values were then taken to obtain the Kuramoto order parameter, corresponding to the amplitude envelope of local oscillations. In this model, a “local” control parameter K determines the homogeneous coupling strength within the nodes while a “global” control parameter L determines the inter-regional coupling strength based on the pairwise-connectivity white-matter axonal fiber counts estimated with structural diffusion tensor imaging (DTI) and obtained from the Human Connectome Project (HCP).

Also, I was wondering why there is no structural disorder or across-compartment variability in the model. Given that what the authors aim at studying is precisely the possible variable distance to criticality across brain regions, these should be included in a model, even if it is parsimonious.

- **(R)** Structural disorder in the model is governed by the structural connectome (SC) for which we illustrate two levels of heterogeneity: realistic and uniform. The structural connectome connects the nodes and is scaled by L , and thus represents 'across-compartment variability'. Within the nodes, the connectivity among oscillators is always uniform to reduce computational load and improve conceptual clarity as the topological structure of local neuronal networks.
- **(A)** This aspect of the model is now clarified in the new Fig. 1a-c and the corresponding Introduction, Methods (see above) and Results text sections. The new Fig. 1b and c also illustrate how the distance from criticality is reflected in the relationship of synchronization and LRTCs.

Results, p. 6, line 24 – p.7, line 4:

A “within-node” control parameter K was used to scale the uniform, all-to-all coupling among the oscillators in each node, while an independent “inter-node” control parameter L scaled the structural connectivity between the nodes. We first simulated both classical brain criticality and critical-like GP by varying the heterogeneity of structural connectivity (see “Parameter space” in Fig. 1a). To this end, for classical brain criticality, all nodes were coupled with uniform connectivity, while for the GP, the coupling between nodes was defined with the realistic and highly heterogenous structural connectome of the human cerebral white-matter tracts.

It would be very illuminating to see the model, actually having diverse LRTC at different nodes.

- **(R)** This is a very important aspect. The present model, in fact, achieves diverse LRTCs across nodes and in a manner that is similar with the real observed cortical architecture and diversity of LRTCs. This fact is illustrated above in *Figure for reviewers R5* and based on a manuscript-in-preparation of ours.

Related to this, it would also be nice to see the Griffith phase (GP) in the model. If empirical data (allegedly lying in a GP) are to be compared with a model, at least the model should have a GP and it should be quantified, somehow.

- **(R)** We thank the reviewer for this suggestion and agree that this is central for the manuscript.
- **(A)** We have now fully revised the computational modeling sections to explicitly demonstrate the stretching of the classical critical point into a GP upon the transition from uniform structural connectivity to the realistic, highly heterogenous one (please see new figure Fig. 1b). We also introduce here the conceptual illustration (see Fig. 1a) on how heterogeneity is a “structural control parameter” regulating the width of GP. Finally, as a novel observation, we illustrate (Fig. 1b) how structural heterogeneity not only expands the critical point into a GP but also greatly amplifies

the dynamic heterogeneity arising from a fixed range of inter-subject structural variability.

In summary: I see the modelling effort as exceedingly naive and not properly corresponding to the empirical measurements/findings, so it turns out to be of very little utility.

- **(R)** We politely disagree with this notion and hope that the many improvements in the present manuscript clarify the utility value of our modeling efforts.
- **(R)** The Kuramoto model is a universal standard model for synchronization dynamics and has been widely applied also to understanding the dynamics of neural oscillations and synchronization. Despite its simplicity, it is capable of capturing the essence of rich neuronal dynamics including mesoscopic (Breakspear et al., 2010) and macroscopic (Roberts et al., 2019) complex spatiotemporal brain waves, a wide array of bifurcation types (Acebrón et al., 2005) that are analytically tractable (Martens et al., 2009) similarly to many other more biophysically detailed models. Importantly, the Kuramoto model was recently shown to produce Griffiths-phase like frustrated synchronization dynamics in large synthetic hierarchical networks (Buendía et al., 2020, 2021). Therefore, in our opinion, it is a relevant and potent dynamical model, especially when combined with realistic human structural connectome data to enable the modeling of the Griffiths phase. We also would like to emphasize that “synchronization dynamics” is specifically the construct of interest in the present study – it is comprehensively modeled by the Kuramoto model and it remains unclear what advantages would more complex models yield and whether these would outweigh the disadvantages of their greatly expanded parameter spaces.
- **(R)** Finally, we would like to point out that the model is aimed to reproduce brain dynamics as recorded with electrophysiological methods. We have successfully used the model to produce accurately brain dynamics also in previous articles (Siebenhühner et al., 2020; Simola et al., 2022) and our findings are well-aligned with the most recent theoretical work on GP in large-scale hierarchical-modular networks (Buendía et al. 2021; 2022). Therefore, we are confident that the model is able to capture several essential aspects of experimentally observable brain dynamics.

2) Dependence of frequency band:

I find it extremely obscure and confusing all the discussion on how results depend on the considered frequency band: some effects such as large synchronization or strong correlations appear in some bands but not in others, and these change from one quantity to another.

At the end of the day I remain strongly confused about what is the role of criticality (and the distance to criticality) in each band? For instance, I think the reader can have a hard time trying to make sense

of the peaks and valleys in the second row of Fig.2 (or the first and third in Fig.3). Are some there bands closer to critical to others? which ones? Are there bands that are definitely not critical?

- **(R)** The notion of GP being potentially variable across frequencies is indeed a novel finding. However, several lines of prior art (at least tens of publications both from our and many other labs) have repeatedly shown great variability in the strength of LRTCs across frequency bands. As LRTCs are quadratically dependent on the proximity to a critical point, these prior art already imply that not all oscillations are “equally critical”. Moreover, the apparent lack of significant LRTCs in some bands suggests that neuronal assemblies might indeed not be critical at all under the studied conditions, but it is important to note that this is a methodologically challenging question and difficult to test rigorously. To the best of our knowledge, there are no publications reporting rigorous negative evidence for lack of critical-like dynamics in a specific frequency band, and several of our studies do find evidence of significant LRTCs throughout the 1-300 Hz frequency range.
- **(R)** From the neurobiological perspective, variability across frequencies is not surprising as oscillations in different frequencies are driven by different underlying neurophysiological mechanisms and appear primarily in distinct functional networks that, e.g., are subject to variable neuromodulation.
- **(A)** We have now clarified these items in the Discussion.

Discussion p. 12, line 19 – 28:

These findings strongly suggest that individual and regional variability in the synchronization levels is explained by the variability in the operating points in line with the GP framework. The GP framework thus extends the classical notion of criticality being constrained to a singular point in the space of control parameters which underlies models in which the brain is assumed to operate either at the critical point (Beggs & Plenz, 2003), near the critical point (Chialvo, 2010), or in a slightly subcritical regime (Priesemann et al., 2014; Toker et al., 2022; Wilting & Priesemann, 2019). Importantly, both here and in our earlier study (Zhigalov et al., 2017), we found heterogeneity in dynamics as well as co-variance of LRCTs and synchronization levels also across neocortical regions within specific frequency bands. This suggests that the different functional brain systems in individual subjects operate in different positions of the critical regime (Zhigalov et al., 2017) rather than being fully controlled by a global parameter.

Discussion, p. 14, lines 1–21:

We observed variability in the strength of correlations not only across brain regions but also across frequencies, indicating that brain functional networks have a small variability in their operating points in line with the GP framework where the GP arises due to functional and structural heterogeneity (Muñoz et al., 2010). The correlations were linearly positive at the whole brain level and in most parcels and frequencies. However, in alpha, beta, and high-gamma bands there was also a quadratic trend, showing that in a subset of subjects, these functional networks operated around the “peak” of the extended critical regime. Given the fundamental functional roles that alpha and gamma-band oscillations are thought to play in feed-back and feed-forward information processing (Bastos et al., 2015; Jensen et al., 2015; S. Palva & Palva, 2007, 2011; Samaha et al., 2020), respectively, it is conceivable that these oscillations operate around the peak of the putative critical regime,

enabling maximal dynamic range and flexibility. The localization of strongest alpha-band correlations into dorsal visual areas and PFC and gamma-band correlations into the ventral stream areas is in line with previous findings of the differential localization of power and synchronization in these frequencies (Honkanen et al., 2015; Klimesch et al., 2007; Popov et al., 2018; Singer, 1999) and with their biologically differential dependence on anatomy, neurobiological mechanisms (Simola et al., 2022; Vezoli et al., 2021; Vinck & Perrenoud, 2019), and neuromodulatory genes. These findings of heterogeneous neuronal dynamics are also in agreement with previous study from awake mouse visual cortex where scale-free neural activity was limited to specific subsets of neurons (Jones et al., 2023). Heterogeneity across frequencies and functional neuroanatomy could provide a means for adapting behaviour according to environmental demands.

The computational model does not help at all to clarify this as there is a well defined frequency interval, or not? can the model be extended to describe broadband signals? In summary, all analysis and discussion on the band-frequency dependence should be profoundly clarified.

- **(R)** We agree that the description of the computational model in the previous version of the manuscript was not detailed enough. The model simulates narrow-band population oscillations arising both from the local interactions of phase oscillators and inter-areal interactions of the nodes composed of these oscillators. As such, the model quantifies universal synchronization dynamics, which was the objective of the present study. The model does not have conduction delays between nodes and hence there is no ‘grounding’ of the model to specific time scales or real-world frequencies. While the model central frequency is thus ‘arbitrary’, qualitatively identical dynamics are observable also in models with conduction delays.
- **(R)** We focused on neuronal oscillations, not only because they are fundamental biological mechanisms associated with temporal coordination and communication of information (Fries, 2015; Singer, 1999), but also because they arise from bidirectional interactions between regions. This is crucial as the GP and brain criticality hypothesis specifically yield predictions about the interactions arising between nodes due to heterogeneity of structural connections. Such bidirectional interactions are not similarly defined for broad-band activity and therefore they are not in the focus of this study.
- **(A)** We have now clarified the description of the computational model in Methods (p.18, line 17 – p.19, line 19) and Results (p.6, line 19 – p.7, line 24).

The measurement of synchronization is rather naive and neglects, e.g., the fact that synchronization can occur in a delayed way (e.g. if there are travelling waves). How do enhanced estimators of synchrony affect the results?

- **(R)** We thank the reviewer for pointing this out. We would like to emphasize first the fact that synchronization and traveling waves are distinct constructs even though both may be operationalized with a measurement of the consistency of phase relationships. They also are likely to be achieved via distinct neuronal and systems-level mechanisms.

Per one definition, traveling waves arise from unidirectional interactions between two (or many short-range-connected) nodes, where the nodes entrain the neuronal activity 'sequentially' in a propagating cascade. In contrast, synchronization is thought to arise as a bidirectional interaction between the two nodes (Lakatos et al., 2019).

In terms of oscillations, the brain criticality framework has so far been considered in the context of coupled oscillators and their synchronization. We are not aware of a single paper relating the properties of traveling waves with criticality (outside the scope of avalanche dynamics, as in (Palva et al., 2013; Zhigalov et al., 2015, 2017).

- **(R)** However, PLV and wPLI – which are used to measure synchronization in this study – only measure statistical relationships between timeseries. Hence, these measures are agnostic to delay and also sensitive to delayed interactions.
- **(A)** As per reviewer's suggestion, we have now, nevertheless, computed additional analyses of synchronization in which we added a lag of 1 or 2 cycles between the two time series. These results are shown in new Supplementary Figure 5. As expected the results are similar to the findings without added lag as presented in Figure 2.
- **(R)** Traveling waves in human neuronal data are very difficult to dissociate from synchronization. This is because long-time averaged synchrony is known to decay over distance both in densely (Dotson et al., 2017) and in sparsely (Arnulfo et al., 2020) sampled invasive brain data. Phase difference generally increases with (geodesic and/or WM tract) distance due to conduction delays. Recent study shows that traveling waves identified in MEG/EEG can be explained by mixing at the scalp level equally well as by a cortical travelling wave. See *Zhigalov & Jensen, preprint, <https://doi.org/10.1101/2022.09.28.509870>*.
- **(R)** Given that the brain criticality hypothesis does not make any predictions of the traveling waves and that traveling waves are difficult to dissociate from the source mixing, we would respectfully consider the traveling waves to be outside of the scope and rationale of the present study and we have not included traveling waves analyses in the present study.

3) Also, the way to infer the distance to criticality --by estimating the degree of quadratic correlation after removing linear ones-- seems not very accurate/convincing to me. In particular the plots in the third row of Fig.3. are a mess from which overall trends are barely discernible.

I think the reader would be much more convinced if a single number assigning a distance to criticality to each individual (or brain region?) could be assigned (other studies succeed at doing so).

- **(R)** We assume that the reviewer is referring to scatter plots of quadratic correlations (now shown in Figure 2e,f). We would like to point out that all statistical data providing evidence for the hypothesis that individual synchronization levels are

predicted by the brain criticality are presented in Figure 2c,d; whereas Fig 2e,f are merely meant to illustrate the effect.

- **(R)** We respectfully disagree with the reviewer here that the presence of quadratic correlations could not be used as evidence for the brain operating at the critical regime. The findings from modeling indicate clearly that and why correlations should be quadratic at around the critical point (Figure 1a-c). We apologize however, if this was not explained and represented well in the previous version of the manuscript.
- **(A)** We now have revised our figures and the Results section in order to better clarify the hypothesis.
- **(R)** Quantifying distance from the critical point would indeed be important. There are some attempts such as fE/I (see Bruining et al., 2020) and personalized β_p based on Ising model (Ruffini et al., 2023). However, these approaches all have their drawbacks and are by no means established. We are not aware of any commonly established method for assigning a single number for brain criticality. Finding a better method is no doubt a worthy goal, but we consider this a task for future study. We would like to hence stick here with a concrete and established framework of brain criticality to measure whether brain criticality predicts individual synchronization levels at the group level using the large cohort of subjects available for this study. Meanwhile, we report evidence in this study using group-level statistics, as is usual in the field.

4) I personally dislike calling "connectivity strength" to the level of nodal or global synchronization. Actual (physical) connectivity is one thing and the level of correlation/synchronization is another. Of course, this is a semantic issue, but I believe many readers would share my confusion with the use of connectivity to refer to synchronization.

- **(A)** We agree that "connectivity" is not an optimal term. We have now completely switched to using the term "synchronization" throughout the manuscript to avoid any possible confusion.

5) In general all plots are hardly illuminating. I do not see "by eye" simple trends that definitely convince me that we are seeing some real trends. This fact, mixed with the confusion between the different bands and the two different types of recordings (that segregate the patients already in two groups and makes the comparison between them doubtful), makes me not feel persuaded at all by the authors conclusions. I think many potential readers will share this sense of dissatisfaction with me.

- **(R)** We are sorry that the reviewer doesn't find the plots illuminating. We would like to clarify here that all conclusions are based on the statistical analyses presented in Figures 2c,d, 3a,c,e,g and 4e,f. These figures do not show the results as "trends"

but as correlations, which show clearly the presence of statistically robust linear and quadratic correlations.

- **(R)** We believe that our analysis of healthy brain activity is strengthened by using both MEG data from healthy subjects and SEEG data from healthy brain areas in epileptics. This is because SEEG is considered as a gold standard for electrophysiological data as it records directly neuronal activity from brains and hence does not suffer from source leakage and signal smearing as MEG. Hence, reporting similar findings with both MEG and SEEG strengthens our results in our opinion.
- **(R)** It is true that the results are slightly different between MEG data and healthy non-EZ contacts in SEEG data. In SEEG, there were less quadratic correlations in comparison to MEG data suggesting that in SEEG neuronal activity was more on the subcritical side. It should be noted that besides having two different groups of subjects, SEEG and MEG measure activity from neuronal populations of different sizes. Nevertheless, we believe that the results are similar enough to support our conclusions about healthy brain activity.
- **(R)** The major benefit of the SEEG data in the present paper, however, is that it allows us to view on epileptic activity in EZ contacts which is known to originate from increased excitability. By using these data, we were able to test the hypothesis that increased excitation shifts the system to the supercritical side of the extended critical regime which is confirmed by the presence of negative linear correlations.

6) I do not see how Heiney et al. is an adequate reference for the existence of LRTC in vitro. On the other hand, I would humbly suggest that the authors read and cite the recent review paper: O'Byrne, J., & Jerbi, K. (2022). Trends in Neurosciences that is very pertinent to the issues addressed here.

- **(A)** We thank the reviewer for pointing us towards the paper by O'Byrne and Jerbi which we are now citing in our manuscript, and agree that Heiney et al. is not an adequate reference for LRTCs in vitro. We are also now citing (Yang et al., 2012) in support of our claims.

Discussion p. 14, lines 23 – 27:

Brain criticality is primarily thought to be controlled by the finely balanced E/I ratio, excessive excitation may lead to operation in the super-critical phase as has been shown in in vitro (Plenz & Thiagarajan, 2007; Toker et al., 2022; Yang et al., 2012) and modelling (Poil et al., 2012) studies. We thus investigated whether inter-ictal brain dynamics in epileptogenic brain regions would be characterised by supercritical-like dynamics.

In summary, I am not satisfied with the methodologies, analyses performed and clarity of the conclusions so, thus far, I cannot recommend publication of the manuscript in a high-quality journal such as Nat. Comm. At least not in its present form.

- **(R)** We thank the reviewer for their honest assessment that has guided us and hope that they will find the revised manuscript more satisfactory.

References

- Acebrón, J. A., Bonilla, L. L., Pérez Vicente, C. J., Ritort, F., & Spigler, R. (2005). The Kuramoto model: A simple paradigm for synchronization phenomena. *Reviews of Modern Physics*, 77(1), 137–185. <https://doi.org/10.1103/RevModPhys.77.137>
- Arnulfo, G., Wang, S. H., Myrov, V., Toselli, B., Hirvonen, J., Fato, M. M., Nobili, L., Cardinale, F., Rubino, A., Zhigalov, A., Palva, S., & Palva, J. M. (2020). Long-range phase synchronization of high-frequency oscillations in human cortex. *Nature Communications*, 11(1). <https://doi.org/10.1038/s41467-020-18975-8>
- Breakspear, M., Heitmann, S., & Daffertshofer, A. (2010). Generative Models of Cortical Oscillations: Neurobiological Implications of the Kuramoto Model. *Frontiers in Human Neuroscience*, 4. <https://doi.org/10.3389/fnhum.2010.00190>
- Bruining, H., Hardstone, R., Juarez-Martinez, E. L., Sprengers, J., Avramiea, A.-E., Simpraga, S., Houtman, S. J., Poil, S.-S., Dallares, E., Palva, S., Oranje, B., Palva, J. M., Mansvelder, H. D., & Linkenkaer-Hansen, K. (2020). Measurement of excitation-inhibition ratio in autism spectrum disorder using critical brain dynamics. *Scientific Reports*, 10(1), 9195. <https://doi.org/10.1038/s41598-020-65500-4>
- Buendía, V., di Santo, S., Bonachela, J. A., & Muñoz, M. A. (2020). Feedback Mechanisms for Self-Organization to the Edge of a Phase Transition. In *Frontiers in Physics* (Vol. 8). Frontiers Media S.A. <https://doi.org/10.3389/fphy.2020.00333>
- Buendía, V., Villegas, P., Burioni, R., & Muñoz, M. A. (2021). Hybrid-type synchronization transitions: where marginal coherence, scale-free avalanches, and bistability live together. *Phys. Rev. Research*, 3, 023224. <https://doi.org/10.1103/PhysRevResearch.3.023224>
- Dotson, N. M., Hoffman, S. J., Goodell, B., & Gray, C. M. (2017). A Large-Scale Semi-Chronic Microdrive Recording System for Non-Human Primates. *Neuron*, 96(4), 769-782.e2. <https://doi.org/10.1016/j.neuron.2017.09.050>
- Fries, P. (2015). Rhythms for Cognition: Communication through Coherence. In *Neuron* (Vol. 88, Issue 1, pp. 220–235). Cell Press. <https://doi.org/10.1016/j.neuron.2015.09.034>
- Hardstone, R., Poil, S.-S., Schiavone, G., Jansen, R., Nikulin, V., Mansvelder, H., & Linkenkaer-Hansen, K. (2012). Detrended Fluctuation Analysis: A Scale-Free View

- on Neuronal Oscillations. *Frontiers in Physiology*, 3, 450.
<https://doi.org/10.3389/fphys.2012.00450>
- Lakatos, P., Gross, J., & Thut, G. (2019). A New Unifying Account of the Roles of Neuronal Entrainment. *Current Biology*, 29(18), R890–R905.
<https://doi.org/10.1016/j.cub.2019.07.075>
- Martens, E. A., Barreto, E., Strogatz, S. H., Ott, E., So, P., & Antonsen, T. M. (2009). Exact results for the Kuramoto model with a bimodal frequency distribution. *Physical Review E*, 79(2), 026204. <https://doi.org/10.1103/PhysRevE.79.026204>
- Meisel, C., Schulze-Bonhage, A., Freestone, D., Cook, M. J., Achermann, P., & Plenz, D. (2015). Intrinsic excitability measures track antiepileptic drug action and uncover increasing/decreasing excitability over the wake/sleep cycle. *Proceedings of the National Academy of Sciences*, 112(47), 14694–14699.
<https://doi.org/10.1073/pnas.1513716112>
- Müller, P. M., & Meisel, C. (2023). Spatial and temporal correlations in human cortex are inherently linked and predicted by functional hierarchy, vigilance state as well as antiepileptic drug load. *PLOS Computational Biology*, 19(3), e1010919.
<https://doi.org/10.1371/journal.pcbi.1010919>
- Muñoz, M. A., Juhász, R., Castellano, C., & Ódor, G. (2010). Griffiths Phases on Complex Networks. *Phys. Rev. Lett.*, 105(12), 128701.
<https://doi.org/10.1103/PhysRevLett.105.128701>
- Palva, J. M., Wang, S. H., Palva, S., Zhigalov, A., Monto, S., Brookes, M. J., Schoffelen, J. M., & Jerbi, K. (2018). Ghost interactions in MEG/EEG source space: A note of caution on inter-areal coupling measures. *NeuroImage*, 173, 632–643.
<https://doi.org/10.1016/j.neuroimage.2018.02.032>
- Palva, J. M., Zhigalov, A., Hirvonen, J., Korhonen, O., Linkenkaer-Hansen, K., & Palva, S. (2013). Neuronal long-range temporal correlations and avalanche dynamics are correlated with behavioral scaling laws. *Proceedings of the National Academy of Sciences of the United States of America*, 110(9), 3585–3590.
<https://doi.org/10.1073/pnas.1216855110>
- Roberts, J. A., Gollo, L. L., Abey Suriya, R. G., Roberts, G., Mitchell, P. B., Woolrich, M. W., & Breakspear, M. (2019). Metastable brain waves. *Nature Communications*, 10(1), 1056. <https://doi.org/10.1038/s41467-019-08999-0>
- Ruffini, G., Damiani, G., Lozano-Soldevilla, D., Deco, N., Rosas, F. E., Kiani, N. A., Ponce-Alvarez, A., Kringelbach, M. L., Carhart-Harris, R., & Deco, G. (2023). LSD-induced increase of Ising temperature and algorithmic complexity of brain dynamics. *PLOS Computational Biology*, 19(2), e1010811.
<https://doi.org/10.1371/journal.pcbi.1010811>
- Siebenhühner, F., Wang, S. H., Arnulfo, G., Lampinen, A., Nobili, L., Palva, J. M., & Palva, S. (2020). Genuine cross-frequency coupling networks in human resting-state electrophysiological recordings. *PLoS Biology*, 18(5).
<https://doi.org/10.1371/journal.pbio.3000685>
- Simola, J., Siebenhühner, F., Myrov, V., Kantojärvi, K., Paunio, T., Palva, J. M., Brattico, E., & Palva, S. (2022). Genetic polymorphisms in COMT and BDNF influence

- synchronization dynamics of human neuronal oscillations. *Science*, 25(9), 104985. <https://doi.org/10.1016/j.isci.2022.104985>
- Singer, W. (1999). Neuronal Synchrony: A Versatile Code Review for the Definition of Relations? *Neuron*, 24, 49–65.
- Yang, H., Shew, W. L., Roy, R., & Plenz, D. (2012). Maximal Variability of Phase Synchrony in Cortical Networks with Neuronal Avalanches. *Journal of Neuroscience*, 32(3), 1061–1072. <https://doi.org/10.1523/JNEUROSCI.2771-11.2012>
- Zhigalov, A., Arnulfo, G., Nobili, L., Palva, S., & Palva, J. M. (2015). Relationship of Fast- and Slow-Timescale Neuronal Dynamics in Human MEG and SEEG. *Journal of Neuroscience*, 35(13), 5385–5396. <https://doi.org/10.1523/JNEUROSCI.4880-14.2015>
- Zhigalov, A., Arnulfo, G., Nobili, L., Palva, S., & Palva, J. M. (2017). Modular co-organization of functional connectivity and scale-free dynamics in the human brain. *Network Neuroscience*, 1(2), 143-165. https://doi.org/10.1162/netn_a_00008

Reviewer #1 (Remarks to the Author):

The authors have addressed my concerns.

Reviewer #2 (Remarks to the Author):

The authors have invested a lot of work for answering all my questions in a reasonably convincing way and for improving this manuscript. Readability is much superior and will enable non-specialists to follow their methods and line of arguments. From my side there are only minor points left which I would ask the authors to correct. There is a general issue with nouns in plural form and verbs conjugated for singular, e.g. the example cited below "dynamics ... is... ". Please carefully check this all throughout the text.

minor

Page 3, Line 4: dynamics = plural: Healthy brain dynamics ARE near critical, but mostly on the subcritical side. Likewise in Line 5 "on the supercritical side..."

Page 12, Line 6: If variability of in synchronization levels would be explained

Reviewer #3 (Remarks to the Author):

I sincerely congratulate the authors for the amazing job they have done in improving the paper following the suggestions of all the reviewers and accommodating for a huge number of suggestions. As a result, in my humble opinion, the quality of the manuscript has increased enormously and I think it is (almost) ready for acceptance. There are still some aspects of the paper that I, personally, would like to tune a bit, but I dislike being exceedingly nagging as a referee, so I refrain myself for asking for further fine-detail refinements.

The only point I would like to raise here is that re-reading the manuscript and looking again into the literature, I realize that the work of Villegas et al. could/should be properly acknowledged.

In particular,

Villegas, P. et al. (2014). Frustrated hierarchical synchronization and emergent complexity in the human connectome network. *Scientific reports*, 4(1), 1-7.

presents the same model (Kuramoto on top of human connectome networks) and seems to be the first reference where Griffith phases are studied in the context of synchronisation problems.

And, in

Di Santo, S., Villegas, P., Burioni, R., & Munoz, M. A. (2018). Landau-Ginzburg theory of cortex dynamics: Scale-free avalanches emerge at the edge of synchronization. *Proceedings of the National Academy of Sciences*, 115(7), E1356-E1365.

there is a measurement of Hurst exponents reporting that they peak at the synchronisation critical point.

Point-by-point response to reviewers' comments

We are sincerely grateful to all three reviewers for their expert feedback on the manuscript, which has helped considerably with improving the work.

Reviewer #1 (Remarks to the Author): The authors have addressed my concerns.

We thank the reviewer for their assessment and positive evaluation of the manuscript.

Reviewer #2 (Remarks to the Author):

The authors have invested a lot of work for answering all my questions in a reasonably convincing way and for improving this manuscript. Readability is much superior and will enable non-specialists to follow their methods and line of arguments. From my side there are only minor points left which I would ask the authors to correct.

We thank the reviewer for their kind words and important insight throughout the review process.

There is a general issue with nouns in plural form and verbs conjugated for singular, e.g. the example cited below "dynamics ... is... ". Please carefully check this all throughout the text.

minor

Page 3, Line 4: dynamics = plural: Healthy brain dynamics ARE near critical, but mostly on the subcritical side. Likewise in Line 5 "on the supercritical side..."

Thank you. We have now corrected this and changed the conjugation of verbs following the term "dynamics".

Page 12, Line 6: If variability of in synchronization levels would be explained

Thank you. We have now corrected this.

Reviewer #3 (Remarks to the Author):

I sincerely congratulate the authors for the amazing job they have done in improving the paper following the suggestions of all the reviewers and accomodating for a huge number of suggestions.

As a result, in my humble opinion, the quality of the manuscript has increased enormously and I think it is (almost) ready for acceptance. There are still some aspects of the paper that I, personally, would like to tune a bit, but I dislike being exceedingly nagging as a referee, so I refrain myself for asking for further fine-detail refinements.

We thank the reviewer for their kind words as well as for the prior comments that helped us a great deal in improving the manuscript.

The only point I would like to raise here is that re-reading the manuscript and looking again into the literature, I realize that the work of Villegas et al. could/should be properly acknowledged.

In particular,

Villegas, P. et al. (2014). Frustrated hierarchical synchronization and emergent complexity in the human connectome network. *Scientific reports*, 4(1), 1-7.

presents the same model (Kuramoto on top of human connectome networks) and seems to be the first reference where Griffith phases are studied in the context of synchronisation problems.

And, in

Di Santo, S., Villegas, P., Burioni, R., & Munoz, M. A. (2018). Landau–Ginzburg theory of cortex dynamics: Scale-free avalanches emerge at the edge of synchronization. *Proceedings of the National Academy of Sciences*, 115(7), E1356-E1365.

there is a measurement of Hurst exponents reporting that they peak at the synchronisation critical point.

We thank the reviewer for these suggestions, and we now cite the two recommend papers. We fully agree that they are central literature in the field and very relevant here.